# GLOBAL CONVERGENCE OF POLICY GRADIENT METHODS FOR LINEARIZED CONTROL PROBLEMS

## ABSTRACT

Direct policy gradient methods for reinforcement learning and continuous control problems are a popular approach for a variety of reasons: 1) they are easy to implement without explicit knowledge of the underlying model 2) they are an "end-to-end" approach, directly optimizing the performance metric of interest 3) they inherently allow for richly parameterized policies. A notable drawback is that even in the most basic continuous control problem (that of linear quadratic regulators), these methods must solve a non-convex optimization problem, where little is understood about their efficiency from both computational and statistical perspectives. In contrast, system identification and model based planning in optimal control theory have a much more solid theoretical footing, where much is known with regards to their computational and statistical properties. This work bridges this gap showing that (model free) policy gradient methods globally converge to the optimal solution and are efficient (polynomially so in relevant problem dependent quantities) with regards to their sample and computational complexities.

## 1 INTRODUCTION

Recent years have seen major advances in the control of uncertain dynamical systems using reinforcement learning and data-driven approaches; examples range from allowing robots to perform more sophisticated controls tasks such as robotic hand manipulation (Tassa et al., 2012; Al Borno et al., 2013; Kumar et al., 2016; Levine et al., 2016; Tobin et al., 2017; Rajeswaran et al., 2017a), to sequential decision making in game domains, e.g. AlphaGo (Silver et al., 2016) and Atari game playing (Mnih et al., 2015). Deep reinforcement learning (DeepRL) are becoming increasingly popular for tackling such challenging sequential decision making problems.

Many of these successes have relied on sampling based reinforcement learning algorithms such as policy gradient methods, including the DeepRL approaches; here, there is little theoretical understanding of their efficiency, either from a statistical or a computational perspective. In contrast, control theory (optimal and adaptive control) has a rich body of tools, with provable guarantees, for related sequential decision making problems, particularly those that involve continuous control. These latter techniques are often model-based — they estimate an explicit dynamical model first (e.g. system identification) and then design optimal controllers.

This work builds bridges between these two lines of work, namely, between optimal control theory and sample based reinforcement learning methods, using ideas from mathematical optimization.

### 1.1 THE OPTIMAL CONTROL PROBLEM

In the standard optimal control problem, the dynamics model $f_t$, where $f_t$ is specified as

$$x_{t+1} = f_t(x_t, u_t, w_t),$$

maps a state $x_t \in \mathbb{R}^d$, a control (the action) $u_t \in \mathbb{R}^k$, and a disturbance $w_t$, to the next state $x_{t+1} \in \mathbb{R}^d$. The objective is to find the control input $u_t$ which minimizes the long term cost,

$$\begin{aligned} \text{minimize} \quad & \sum_{t=1}^{T} c_t(x_t, u_t) \\ \text{such that} \quad & x_{t+1} = f_t(x_t, u_t, w_t). \end{aligned}$$

Here the $u_t$ are allowed to depend on the history of observed states.

In practice, this is often solved by considering the linearized control (sub-)problem where the dynamics are approximated by

$$x_{t+1} = A_t x_t + B_t u_t + w_t,$$

and the costs are approximated by a quadratic function in $x_t$ and $u_t$, e.g. (Todorov & Li, 2004). This work considers an important special case: the time homogenous, infinite horizon problem referred to as the linear quadratic regulator (LQR) problem. The results herein can also be extended to the finite horizon, time in-homogenous setting, discussed in Section 5.

In the LQR problem, the objective is

$$\text{minimize} \quad \mathbb{E} \left[ \sum_{t=0}^{\infty} (x_t^\top Q x_t + u_t^\top R u_t) \right]$$
$$\text{such that} \quad x_{t+1} = A x_t + B u_t, \quad x_0 \sim \mathcal{D}.$$

where initial state $x_0 \sim \mathcal{D}$ is assumed to be randomly distributed according to distribution $\mathcal{D}$; the matrices $A \in \mathbb{R}^{d \times d}$ and $B \in \mathbb{R}^{d \times k}$ are referred to as system (or transition) matrices; $Q \in \mathbb{R}^{d \times d}$ and $R \in \mathbb{R}^{k \times k}$ are both positive definite matrices that parameterize the quadratic costs. For clarity, this work does not consider a noise disturbance but only a random initial state. The importance of (some) randomization for analyzing direct methods is discussed in Section 3.

Throughout, assume that $A$ and $B$ are such that the optimal cost is finite (for example, the controllability of the pair $(A, B)$ would ensure this). Optimal control theory (Anderson & Moore, 1990; Evans, 2005; Bertsekas, 2011; 2017) shows that the optimal control input can be written as a linear function in the state,

$$u_t = -K^* x_t$$

where $K^* \in \mathbb{R}^{k \times d}$.

**Planning with a known model.** Planning can be achieved by solving the algebraic Riccati equation,

$$P = A^T P A + Q - A^T P B (B^T P B + R)^{-1} B^T P A, \tag{1}$$

for a positive definite matrix $P$ which parameterizes the "cost-to-go" (the optimal cost from a state going forward). The optimal control gain is then given as:

$$K^* = -(B^T P B + R)^{-1} B^T P A. \tag{2}$$

There are both algebraic solution methods to find $P$ and (convex) SDP formulations to solve for $P$. More broadly, even though there are convex formulations for planning, these formulations: 1) do not directly parameterize the policy 2) they are not "end-to-end" approaches in that they are not directly optimizing the cost function of interest and 3) it is not immediately clear how to utilize these approaches in the model-free setting, where the agent only has simulation access. These formulations are discussed in Section A, where there is a discussion of how the standard SDP formulation is *not* a direct method that minizes the cost over the set of feasible policies.

## 1.2 CONTRIBUTIONS OF THIS WORK

Even in the most basic case of the standard linear quadratic regulator model, little is understood as to how direct (model-free) policy gradient methods fare. This work provides rigorous guarantees, showing that, while in fact the approach is a non-convex one, directly using (model free) local search methods leads to finding the globally optimal policy. The main contributions are as follows:

- (Exact case) Even with access to exact gradient evaluation, little is understood about whether or not convergence to the optimal policy occurs, even in the limit, due to the non-convexity in the problem. This work shows that global convergence does indeed occur (and does so efficiently) for local search based methods.

- (Model free case) Without a model, this work shows how one can use simulated trajectories (as opposed to having knowledge of the model) in a stochastic policy gradient method where provable convergence to a globally optimal policy is guaranteed, with (polynomially) efficient computational and sample complexities.

- (The natural policy gradient) Natural policy gradient methods (Kakade, 2001) — and related algorithms such as Trust Region Policy Optimization (Schulman et al., 2015) and the natural actor critic (Peters & Schaal, 2007) — are some of the most widely used and effective policy gradient methods (see Duan et al. (2016)). While many results argue in favor of this method based on either information geometry (Kakade, 2001; Bagnell & Schneider, 2003) or based on connections to actor-critic methods (Deisenroth et al., 2013), these results do not provably show an improved convergence rate. This work is the first to provide a guarantee that the natural gradient method enjoys a considerably improved convergence rate over its naive gradient counterpart.

More broadly, the techniques in this work merge ideas from optimal control theory, mathematical (and zeroth order) optimization, and sample based reinforcement learning methods. These techniques may ultimately help in improving upon the existing set of algorithms, addressing issues such as variance reduction or improving upon the natural policy gradient method (with, say, a Gauss-Newton method). The Discussion touches upon some of these issues.

## 1.3 RELATED WORK

In the reinforcement learning setting, the model is unknown, and the agent must learn to act through its interactions with the environment. Here, solution concepts are typically divided into: model-based approaches, where the agent attempts to learn a model of the world, and model-free approaches, where the agent directly learns to act and does not explicitly learn a model of the world. The related work on provably learning LQRs is reviewed from this perspective.

**Model-based learning approaches.** In the context of LQRs, the agent attempts to learn the dynamics of "the plant" (i.e. the model) and then plans, using this model, for control synthesis. Here, the classical approach is to learn the model with subspace identification (Ljung, 1999). Fiechter (1994) provides a provable learning (and non-asymptotic) result, where the quality of the policy obtained is shown to be near optimal (efficiency is in terms of the persistence of the training data and the controllability Gramian). Abbasi-Yadkori & Szepesvári (2011) also provides provable, non-asymptotic learning results (in a regret context), using a bandit algorithm that achieves lower sample complexity (by balancing exploration-exploitation more effectively); the computational efficiency of this approach is less clear.

More recently, Dean et al. (2017) expands on an explicit system identification process, where a robust control synthesis procedure is adopted that relies on a coarse model of the plant matrices ($A$ and $B$ are estimated up to some accuracy level, naturally leading to a "robust control" setup). Arguably, this is the most general (and non-asymptotic) result, that is efficient from both a statistical perspective (computationally, the method works with a finite horizon to approximate the infinite horizon). This result only needs the plant to be controllable; the work herein needs the stronger assumption that the initial policy in the local search procedure is a stable controller (an assumption which may be inherent to local search procedures, discussed in Section 5).

**Model-free learning approaches.** Model-free approaches that do not rely on an explicit system identification step typically either: 1) estimate value functions (or state-action values) through Monte Carlo simulation which are then used in some approximate dynamic programming variant (Bertsekas, 2011) or 2) directly optimize a (parameterized) policy, also through Monte Carlo simulation. Model-free approaches for learning optimal controllers is not well understood, from a theoretical perspective. Here, Bradtke et al. (1994) provides an asymptotic learnability result using a value function approach, namely $Q$-learning.

## 2 PRELIMINARIES AND BACKGROUND

### 2.1 EXACT GRADIENT DESCENT

This work seeks to characterize the behavior of (direct) policy gradient methods, where the policy is linearly parameterized, as specified by a matrix $K \in \mathbb{R}^{k \times d}$ which generates the controls:

$$u_t = -Kx_t$$

for $t \geq 0$. The cost of this $K$ is denoted as:

$$C(K) := \mathbb{E}_{x_0 \sim \mathcal{D}} \left[ \sum_{t=0}^{\infty} (x_t^\top Q x_t + u_t^\top R u_t) \right]$$

where $\{x_t, u_t\}$ is the trajectory induced by following $K$, starting with $x_0 \sim \mathcal{D}$. The importance of (some) randomization, either in $x_0$ or noise through having a disturbance, for analyzing gradient methods is discussed in Section 3. Here, $K^*$ is a minimizer of $C(\cdot)$.

Gradient descent on $C(K)$, with a fixed stepsize $\eta$, follows the update rule:

$$K \leftarrow K - \eta \nabla C(K) .$$

It is helpful to explicitly write out the functional form of the gradient. Define $P_K$ as the solution to:

$$P_K = Q + K^\top R K + (A - BK)^\top P_K (A - BK) .$$

and, under this definition, it follows that $C(K)$ can be written as:

$$C(K) = \mathbb{E}_{x_0 \sim \mathcal{D}} \, x_0^\top P_K x_0 .$$

Also, define $\Sigma_K$ as the (un-normalized) state correlation matrix, i.e.

$$\Sigma_K = \mathbb{E}_{x_0 \sim \mathcal{D}} \sum_{t=0}^{\infty} x_t x_t^\top .$$

**Lemma 1.** *(Policy Gradient Expression) The policy gradient is:*

$$\nabla C(K) = 2 \left( (R + B^\top P_K B) K - B^\top P_K A \right) \Sigma_K$$

*Proof.* Observe:

$$x_0^\top P_K x_0 = x_0^\top \left( Q + K^\top R K \right) x_0 + x_0^\top (A - BK)^\top P_K (A - BK) x_0 .$$

This implies:

$$
\begin{aligned}
x_0^\top \nabla P_K x_0 &= 2 R K x_0 x_0^\top - 2 B^\top P_K (A - BK) x_0 x_0^\top + x_0^\top (A - BK)^\top \nabla P_K (A - BK) x_0 \\
&= 2 \left( (R + B^\top P_K B) K - B^\top P_K A \right) x_0 x_0^\top + x_1^\top \nabla P_K x_1 \\
&= 2 \left( (R + B^\top P_K B) K - B^\top P_K A \right) \sum_{t=0}^{\infty} x_t x_t^\top
\end{aligned}
$$

using recursion and that $x_1 = (A - BK) x_0$. Taking expectations completes the proof. $\square$

## 2.2 Review: (Model free) sample based policy gradient methods

Sample based policy gradient methods introduce some randomization for estimating the gradient.

**REINFORCE.** Let $\pi_\theta(u|x)$ be a parametric stochastic policy, where $u \sim \pi_\theta(\cdot|x)$. The policy gradient of the cost, $C(\theta)$, is:

$$\nabla C(\theta) = \mathbb{E} \left[ \sum_{t=0}^{\infty} Q_{\pi_{\theta_t}}(x_t, u_t) \nabla \log \pi_\theta(u_t|x_t) \right], \text{ where } Q_{\pi_\theta}(x, u) = \mathbb{E} \left[ \sum_{t=0}^{\infty} c_t | x_0 = x, u_0 = u \right],$$

where the expectation is with respect to the trajectory $\{x_t, u_t\}$ induced under the policy $\pi_\theta$ and where $Q_{\pi_\theta}(x, u)$ is referred to as the state-action value. The REINFORCE algorithm uses Monte Carlo estimates of the gradient obtained by simulating $\pi_\theta$.

**The natural policy gradient.** The natural policy gradient (Kakade, 2001) follows the update:

$$\theta \leftarrow \theta - \eta \, G_\theta^{-1} \nabla C(\theta) , \text{ where } G_\theta = \mathbb{E} \left[ \sum_{t=0}^{\infty} \nabla \log \pi_\theta(u_t|x_t) \nabla \log \pi_\theta(u_t|x_t)^\top \right],$$

where $G_\theta$ is the Fisher information matrix. There are numerous succesful related approaches (Peters & Schaal, 2007; Schulman et al., 2015; Duan et al., 2016). An important special case is using a linear policy with additive Gaussian noise (Rajeswaran et al., 2017b), i.e.

$$\pi_K(x, u) = \mathcal{N}(Kx, \sigma^2 I) \tag{3}$$

where $K \in \mathbb{R}^{k \times d}$ and $\sigma^2$ is the noise variance. Here, the natural policy gradient of $K$ (when $\sigma$ is considered fixed) takes the form:

$$K \leftarrow K - \eta \nabla C(K) \Sigma_K^{-1} \tag{4}$$

To see this, one can verify that the Fisher matrix of size $kd \times kd$, which is indexed as $[G_K]_{(i,j),(i',j')}$ where $i, i' \in \{1, \ldots k\}$ and $j, j' \in \{1, \ldots d\}$, has a block diagonal form where the only non-zeros blocks are $[G_K]_{(i,\cdot),(i,\cdot)} = \Sigma_K$ (this is the block corresponding to the $i$-th coordinate of the action, as $i$ ranges from 1 to $k$). This form holds more generally, for any diagonal noise.

**Zeroth order optimization.** Zeroth order optimization is a generic procedure (Conn et al., 2009; Nesterov & Spokoiny, 2015) for optimizing a function $f(x)$, using only query access to the function values of $f(\cdot)$ at input points $x$ (and without explicit query access to the gradients of $f$). This is also the approach in using "evolutionary strategies" (Salimans et al., 2017). The generic approach can be described as follows: define the perturbed function as

$$f_{\sigma^2}(x) = \mathbb{E}_{\varepsilon \sim \mathcal{N}(0,\sigma^2 I)}[f(x + \varepsilon)]$$

For small $\sigma$, the smooth function is a good approximation to the original function. Due to the Gaussian smoothing, the gradient has the particularly simple functional form (see Conn et al. (2009); Nesterov & Spokoiny (2015)):

$$\nabla f_{\sigma^2}(x) = \frac{1}{\sigma^2} \mathbb{E}_{\varepsilon \sim \mathcal{N}(0,\sigma^2 I)}[f(x + \varepsilon)\varepsilon].$$

This expression implies a straightforward method to obtain an unbiased estimate of the $\nabla f_{\sigma^2}(x)$, through obtaining only the function values $f(x + \varepsilon)$ for random $\varepsilon$.

## 3 THE (NON-CONVEX) OPTIMIZATION LANDSCAPE

This section provides a brief characterization of the optimization landscape, in order to help provide intuition as to why global convergence is possible and as to where the analysis difficulties lie.

**Lemma 2.** *(Non-convexity) If $d \geq 3$, there exists an LQR optimization problem, $\min_K C(K)$, which is not convex, quasi-convex, and star-convex.*

Section B provides a specific example. In general, for a non-convex optimization problem, gradient descent may not even converge to the global optima in the limit. For the case of LQRs, the following corollary (of Lemma 8) provides a characterization of the stationary points.

**Corollary 3.** *(Stationary point characterization) If $\nabla C(K) = 0$, then either $K$ is an optimal policy or $\Sigma_K$ is rank deficient.*

This lemma is the motivation for using a distribution over $x_0$ (as opposed to a deterministic starting point): $\mathbb{E}_{x_0 \sim \mathcal{D}} x_0 x_0^\top$ being full rank guarantees that $\Sigma_K$ is full rank, which implies all stationary points are a global optima. An additive disturbance in the dynamics model also suffices.

The concept of gradient domination is important in the non-convex optimization literature (Polyak, 1963; Nesterov & Polyak, 2006; Karimi et al., 2016). A function $f : \mathbb{R}^d \to \mathbb{R}$ is said to be gradient dominated if there exists some constant $\lambda$, such that for all $x$,

$$f(x) - \min_{x'} f(x') \leq \lambda \|\nabla f(x)\|^2.$$

If a function is gradient dominated, this implies that if the magnitude of the gradient is small at some $x$, then the function value at $x$ will be close to that of the optimal function value.

The following corollary of Lemma 8 shows that $C(K)$ is gradient dominated.

**Corollary 4.** *(Gradient Domination) Suppose $\mathbb{E}_{x_0 \sim \mathcal{D}} x_0 x_0^\top$ is full rank. Then $C(K)$ is gradient dominated, i.e.*

$$C(K) - C(K^*) \leq \lambda \langle \nabla C(K), \nabla C(K) \rangle$$

*where $\lambda$ is a problem dependent constant (and $\langle \cdot, \cdot \rangle$ denotes the trace inner product).*

With gradient domination and no (spurious) local optima, one may hope that recent results on escaping saddle points (Nesterov & Polyak, 2006; Ge et al., 2015; Jin et al., 2017) immediately imply that gradient descent converges quickly. This is not the case due to that it is not straightforward to characterize the (local) smoothness properties of $C(K)$; this is a difficulty well studied in the optimal control theory literature, related to robustness and stability. In fact, if it were the case that $C(K)$ is a smooth function[1] (in addition to being gradient dominated), then classical mathematical optimization results (Polyak, 1963) would not only immediately imply global convergence, these results would also imply convergence at a linear rate.

## 4 MAIN RESULTS

First, results on exact gradient methods are provided. From an analysis perspective, this is the natural starting point; once global convergence is established for exact methods, the question of using simulation-based, model-free methods can be approached with zeroth-order optimization methods.

**Notation.** $\|Z\|$ denotes the spectral norm of a matrix $Z$; $\mathrm{Tr}(Z)$ denotes the trace of a square matrix; $\sigma_{\min}(Z)$ denotes the minimal singular value of a square matrix $Z$. Also, it is helpful to define

$$\mu := \sigma_{\min}(\mathbb{E}_{x_0 \sim \mathcal{D}} x_0 x_0^\top)$$

### 4.1 MODEL-BASED OPTIMIZATION: EXACT GRADIENT METHODS

The following three exact update rules are considered:

$$\text{Gradient descent:} \quad K_{n+1} = \quad K_n - \eta \nabla C(K_n) \tag{5}$$

$$\text{Natural policy gradient descent:} \quad K_{n+1} = \quad K_n - \eta \nabla C(K_n) \Sigma_{K_n}^{-1} \tag{6}$$

$$\text{Gauss-Newton:} \quad K_{n+1} = \quad K_n - \eta (R + B^\top P_{K_n} B)^{-1} \nabla C(K_n) \Sigma_{K_n}^{-1}. \tag{7}$$

The natural policy gradient descent direction is defined so that it is consistent with the stochastic case, as per Equation 4. It is straightforward to verify that the policy iteration algorithm is a special case of the Gauss-Newton method when $\eta = 1$ (for the case of policy iteration, convergence in the limit is provided in Todorov & Li (2004); Ng et al. (2002); Liao & Shoemaker (1991), along with local convergence rates.)

The Gauss-Newton method requires the most complex oracle to implement: it requires access to $\nabla C(K)$, $\Sigma_K$, and $R + B^\top P_K B$; it also enjoys the strongest convergence rate guarantee. At the other extreme, gradient descent requires oracle access to only $\nabla C(K)$ and has the slowest convergence rate. The natural policy gradient sits in between, requiring oracle access to $\nabla C(K)$ and $\Sigma_K$, and having a convergence rate between the other two methods.

**Theorem 5.** *(Global Convergence of Gradient Methods) Suppose $C(K_0)$ is finite and and $\mu > 0$.*

- *Gauss-Newton case: For a stepsize $\eta = 1$ and for*

$$N > \frac{\|\Sigma_{K^*}\|}{\mu} \log \frac{C(K_0) - C(K^*)}{\varepsilon},$$

  *the Gauss-Newton algorithm (Equation 7) enjoys the following performance bound:*

$$C(K_N) - C(K^*) \leq \varepsilon$$

- *Natural policy gradient case: For a stepsize*

$$\eta = \frac{1}{\|R\| + \frac{\|B\|^2 C(K_0)}{\mu}}$$

  *and for*

$$N > \frac{\|\Sigma_{K^*}\|}{\mu} \left( \frac{\|R\|}{\sigma_{min}(R)} + \frac{\|B\|^2 C(K_0)}{\mu \sigma_{min}(R)} \right) \log \frac{C(K_0) - C(K^*)}{\varepsilon},$$

  *natural policy gradient descent (Equation 6) enjoys the following performance bound:*

$$C(K_N) - C(K^*) \leq \varepsilon.$$

---

[1] A differentiable function $f(x)$ is said to be smooth if the gradients of $f$ are continuous.

---

**Algorithm 1** Model-Free Policy Gradient (and Natural Policy Gradient) Estimation

---
1: Input: $K$, number of trajectories $m$, roll out length $\ell$, smoothing parameter $r$, dimension $d$
2: **for** $i = 1, \cdots m$ **do**
3:      Sample a policy $\widehat{K}_i = K + U_i$, where $U_i$ is drawn uniformly at random over matrices whose (Frobenius) norm is $r$.
4:      Simulate $\widehat{K}_i$ for $\ell$ steps starting from $x_0 \sim \mathcal{D}$. Let $\widehat{C}_i$ and $\widehat{\Sigma}_i$ be the empirical estimates:

$$\widehat{C}_i = \sum_{t=1}^{\ell} c_t, \quad \widehat{\Sigma}_i = \sum_{t=1}^{\ell} x_t x_t^\top$$

     where $c_t$ and $x_t$ are the costs and states on this trajectory.
5: **end for**
6: Return the (biased) estimates:

$$\widehat{\nabla C(K)} = \frac{1}{m} \sum_{i=1}^{m} \frac{d}{r^2} \widehat{C}_i U_i, \quad \widehat{\Sigma}_K = \frac{1}{m} \sum_{i=1}^{m} \widehat{\Sigma}_i$$

---

- *Gradient descent case: For an appropriate (constant) setting of the stepsize $\eta$,*

$$\eta = \text{poly}\left(\frac{\mu\sigma_{min}(Q)}{C(K_0)}, \frac{1}{\|A\|}, \frac{1}{\|B\|}, \frac{1}{\|R\|}, \sigma_{min}(R)\right)$$

*and for*

$$N \geq \frac{\|\Sigma_{K^*}\|}{\mu} \log \frac{C(K_0) - C(K^*)}{\varepsilon} * \text{poly}\left(\frac{C(K_0)}{\mu\sigma_{min}(Q)}, \|A\|, \|B\|, \|R\|, \frac{1}{\sigma_{min}(R)}\right),$$

*gradient descent (Equation 5) enjoys the following performance bound:*

$$C(K_N) - C(K^*) \leq \varepsilon.$$

In comparison to model-based approaches, these results require the (possibly) stronger assumption that the initial policy is a stable controller, i.e. $C(K_0)$ is finite (an assumption which may be inherent to local search procedures). The Discussion mentions this as direction of future work.

## 4.2 MODEL FREE OPTIMIZATION: SAMPLE BASED POLICY GRADIENT METHODS

In the model free setting, the controller has only simulation access to the model; the model parameters, $A$, $B$, $Q$ and $R$, are unknown. The standard optimal control theory approach is to use system identification to learn the model, and then plan with this learned model This section proves that model-free, policy gradient methods also lead to globally optimal policies, with both polynomial computational and sample complexities (in the relevant quantities).

Using a zeroth-order optimization approach (see Section 2.2), Algorithm 1 provides a procedure to find (controllably biased) estimates, $\widehat{\nabla C(K)}$ and $\widehat{\Sigma}_K$, of both $\nabla C(K)$ and $\Sigma_K$. These can then be used in the policy gradient and natural policy gradient updates as follows:

$$\text{Gradient descent:} \quad K_{n+1} = K_n - \eta\widehat{\nabla C(K_n)} \tag{8}$$

$$\text{Natural policy gradient descent:} \quad K_{n+1} = K_n - \eta\widehat{\nabla C(K_n)}\widehat{\Sigma}_{K_n}^{-1}, \tag{9}$$

where Algorithm 1 is called at every iteration to provide the estimates of $\nabla C(K_n)$ and $\Sigma_{K_n}$.

The choice of using zeroth order optimization vs using REINFORCE (with Gaussian additive noise, as in Equation 3) is primarily for technical reasons[2]. It is plausible that the REINFORCE estimation procedure has lower variance. One additional minor difference, again for technical reasons, is that Algorithm 1 uses a perturbation from the surface of a sphere (as opposed to a Gaussian perturbation).

---

[2]The correlations in the state-action value estimates in REINFORCE are more challenging to analyze.

**Theorem 6.** *(Global Convergence in the Model Free Setting) Suppose $C(K_0)$ is finite, $\mu > 0$, and that $x_0 \sim \mathcal{D}$ has norm bounded by $L$ almost surely. Also, for both the policy gradient method and the natural policy gradient method, suppose Algorithm 1 is called with parameters:*

$$m, \ell, 1/r = poly\left(C(K_0), \frac{1}{\mu}, \frac{1}{\sigma_{min}(Q)}, \|A\|, \|B\|, \|R\|, \frac{1}{\sigma_{min}(R)}, d, 1/\epsilon, L^2/\mu\right) .$$

- *Natural policy gradient case: For a stepsize*

$$\eta = \frac{1}{\|R\| + \frac{\|B\|^2 C(K_0)}{\mu}}$$

*and for*

$$N > \frac{\|\Sigma_{K^*}\|}{\mu} \left(\frac{\|R\|}{\sigma_{min}(R)} + \frac{\|B\|^2 C(K_0)}{\mu \sigma_{min}(R)}\right) \log \frac{2(C(K_0) - C(K^*))}{\varepsilon} ,$$

*then, with high probability, i.e. with probability greater than $1 - \exp(-d)$, the natural policy gradient descent update (Equation 9) enjoys the following performance bound:*

$$C(K_N) - C(K^*) \leq \varepsilon .$$

- *Gradient descent case: For an appropriate (constant) setting of the stepsize $\eta$,*

$$\eta = \text{poly}\left(\frac{\mu \sigma_{min}(Q)}{C(K_0)}, \frac{1}{\|A\|}, \frac{1}{\|B\|}, \frac{1}{\|R\|}, \sigma_{min}(R)\right)$$

*and if $N$ satisfies*

$$N \geq \frac{\|\Sigma_{K^*}\|}{\mu} \log \frac{C(K_0) - C(K^*)}{\varepsilon} * \text{poly}\left(\frac{C(K_0)}{\mu \sigma_{min}(Q)}, \|A\|, \|B\|, \|R\|, \frac{1}{\sigma_{min}(R)}\right) ,$$

*then, with high probability, gradient descent (Equation 8) enjoys the following performance bound:*

$$C(K_N) - C(K^*) \leq \varepsilon .$$

## 5 CONCLUSIONS AND DISCUSSION

This work has provided provable guarantees that model-based gradient methods and model-free (sample based) policy gradient methods convergence to the globally optimal solution, with finite polynomial computational and sample complexities. Taken together, the results herein place these popular and practical policy gradient approaches on a firm theoretical footing, making them comparable to other principled approaches (e.g. subspace ID methods and algebraic iterative approaches).

**Finite $C(K_0)$ assumption, noisy case, and finite horizon case.** These methods allow for extensions to the noisy case and the finite horizon case. This work also made the assumption that $C(K_0)$ is finite, which may not be easy to achieve in some infinite horizon problems. The simplest way to address this is to model the infinite horizon problem with a finite horizon one; the techniques developed in Section D.1 shows this is possible. This is an important direction for future work.

**Open Problems.**

- Variance reduction: This work only proved efficiency from a polynomial sample size perspective. An interesting future direction would be in how to rigorously combine variance reduction methods and model-based methods to further decrease the sample size.

- A sample based Gauss-Newton approach: This work showed how the Gauss-Newton algorithm improves over even the natural policy gradient method, in the exact case. A practically relevant question for the Gauss-Newton method would be how to both: a) construct a sample based estimator b) extend this scheme to deal with (non-linear) parametric policies.

- Robust control: In model based approaches, optimal control theory provides efficient procedures to deal with (bounded) model mis-specification. An important question is how to provably understand robustness in a model free setting.

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

## A   PLANNING WITH A MODEL

This section briefly reviews some parameterizations and solution methods for the classic LQR and related problems from control theory.

**Finite horizon LQR.** First, consider the finite horizon case. The basic approach is to view it as a dynamic program with the value function $x_t^T P_t x_t$, where

$$P_{t-1} = Q + A^T P_k A - A^T P_t B (R + B^T P_t B)^{-1} B^T P_t A,$$

which in turn gives optimal control

$$u_t = -K_t x_t = -(R + B^T P_{t+1} B)^{-1} B^T P_{t+1} A x_t$$

(recursions run backward in time).

Another approach is to view the LQR problem as a linearly-constrained Quadratic Program in all $x_t$ and $u_t$ (where the constraints are given by the dynamics, and the problem size equals the horizon). The QP is clearly a convex problem, but this observation is not useful by itself as the problem size grows with the horizon, and naive use of quadratic programming scales badly. However, the special structure due to linear dynamics allows for simplifications and control-theoretic interpretation as follows: the Lagrange multipliers can be interpreted as "co-state" variables, and they follow a recursion that runs backwards in time known as the "adjoint system". Using Lagrange duality, one can show that this approach is equivalent to solving the Riccati recursion mentioned above.

Popular use of the LQR in control practice is often in the receding horizon LQR, Camacho & Bordons (2004); Rawlings & Mayne (2009): at time $t$, an input sequence is found that minimizes the $T$-step ahead LQR cost starting at the current time, then only the first input in the sequence is used. The resulting static feedback gain converges to the infinite horizon optimal solution as horizon $T$ becomes longer.

**Infinite horizon LQR.** Here, the constrained optimization view (QP) is not informative as the problem is infinite dimensional; the dynamic programming viewpoint readily extends. Suppose the system $A$, $B$ is controllable (which guarantees optimal cost is finite). It turns out that the value function and the optimal controller are static (do not depend on $t$) and can be found by solving the Algebraic Riccati Equation (ARE) given in (1). The optimal $K$ can then be found from equation (2).

The main computational step is solving the ARE, which is extensively studied (e.g. (Lancaster & Rodman, 1995)). One approach due to Kleinman (1968) is to simply run the recursion $P_{k+1} = Q + A^T P_k A - A^T P_k B (R + B^T P_k B)^{-1} B^T P_k A$ with $P_1 = Q$, which converges to the unique positive semidefinite solution of the ARE (since the fixed-point iteration is contractive). Other approaches are direct and based on linear algebra, which carry out an eigenvalue decomposition on a certain block matrix followed by a matrix inversion (Lancaster & Rodman, 1995).

Direct computation of the control input has also been considered in the optimal control literature, e.g., gradient updates in function spaces (Polak, 1973). For the linear quadratic setup, direct iterative computation of the feedback gain has been examined in (Mårtensson & Rantzer, 2009), and explored further in (Mårtensson, 2012) with a view towards distributed implementations. There methods are presented as local search heuristics without provable guarantees of reaching the optimal policy.

**SDP formulation.** The LQR problem can also be expressed as a semidefinite program (SDP) with variable $P$, as given in (Balakrishnan & Vandenberghe, 2003) (section 5, equation (34), this is for a continuous-time system but there are similar discrete-time versions). This SDP can be derived by relaxing the equality in the Riccati equation to an inequality, then using the Schur complement formula to rewrite the resulting Riccati inequality as linear matrix inequality; the objective in the case of LQR is the trace of the positive definite matrix variable. This formulation and its dual has been explored in (Balakrishnan & Vandenberghe, 2003).

It is important to note that while the optimal solution of this SDP is the unique positive semidefinite solution to the Riccati equation, which in turn gives the optimal policy $K^*$, other feasible $P$ (not equal to $P^*$) do not necessarily correspond to a feasible, stabilizing policy $K$. This means that the feasible set of this SDP is not a convex characterization of all $P$ that correspond to stabilizing $K$. Thus it also implies that if one uses any optimization algorithm that maintains iterates in the

feasible set (e.g. interior point methods), no useful policy can be extracted from the iterates before convergence to $P^*$. For this reason, this convex formulation is not helpful for parametrizing the space of policies $K$ in manner that supports the use of local search methods (those that directly lower the cost function of interest), which is the focus of this work.

## B   NON-CONVEXITY OF THE SET OF STABILIZING STATE FEEDBACK GAINS

Let $\mathcal{K}(A, B)$ denote the set of state feedback gains $K$ such that $A - BK$ is stable, i.e., its eigenvalues are inside the unit circle in the complex plane. This set is generally nonconvex. A concise counterexample to convexity is provided here. Let $A$ and $B$ be $3 \times 3$ identity matrices and

$$K_1 = \begin{bmatrix} 1 & 0 & -10 \\ -1 & 1 & 0 \\ 0 & 0 & 1 \end{bmatrix} \quad \text{and} \quad K_2 = \begin{bmatrix} 1 & -10 & 0 \\ 0 & 1 & 0 \\ -1 & 0 & 1 \end{bmatrix}.$$

Then the spectra of $A - BK_1$ and $A - BK_2$ are both concentrated at the origin, yet two of the eigenvalues of $A - B\widehat{K}$ with $\widehat{K} = (K_1 + K_2)/2$, are outside of the unit circle in the complex plane.

## C   ANALYSIS: THE EXACT CASE

This section provides the analysis of the convergence rates of the (exact) gradient based methods. First, some helpful lemmas for the analysis are provided.

Throughout, it is convenient to use the following definition:

$$E_K := (R + B^\top P_K B)K - B^\top P_K A.$$

The policy gradient can then be written as:

$$\nabla C(K) = 2 \left( (R + B^\top P_K B)K - B^\top P_K A \right) \Sigma_K =$$

### C.1   HELPER LEMMAS

Define the value $V_K(x)$, the state-action value $Q_K(x, u)$, and the advantage $A_K(x, u)$. $V_K(x, t)$ is the cost of the policy starting with $x_0 = x$ and proceeding with $K$ onwards:

$$\begin{aligned} V_K(x) & := \sum_{t=0}^{\infty} \left( x_t^\top Q x_t + u_t^\top R u_t \right) \\ & = x^\top P_K x. \end{aligned}$$

$Q_K(x, u)$ is the cost of the policy starting with $x_0 = x$, taking action $u_0 = u$ and then proceeding with $K$ onwards:

$$Q_K(x, u) := x^\top Q x + u^\top R u + V_K(Ax + Bu)$$

The advantage $A_K(x, u)$ is:

$$A_K(x, u) = Q_K(x, u) - V_K(x).$$

The advantage can be viewed as the change in cost starting at state $x$ and taking a one step deviation from the policy $K$.

The next lemma is identical to that in (Kakade & Langford, 2002; Kakade, 2003) for Markov decision processes.

**Lemma 7.** *(Cost difference lemma) Suppose $K$ and $K'$ have finite costs. Let $\{x'_t\}$ and $\{u'_t\}$ be state and action sequences generated by $K'$, i.e. starting with $x'_0 = x$ and using $u'_t = -K'x'_t$. It holds that:*

$$V_{K'}(x) - V_K(x) = \sum_t A_K(x'_t, u'_t).$$

*Also, for any $x$, the advantage is:*

$$A_K(x, K'x) = 2x^\top (K' - K)^\top E_K x + x^\top (K' - K)^\top (R + B^\top P_K B)(K' - K)x. \tag{10}$$

*Proof.* Let $c_t'$ be the cost sequence generated by $K'$. Telescoping the sum appropriately:

$$
\begin{aligned}
V_{K'}(x) - V_K(x) &= \sum_{t=0} c_t' - V_K(x) \\
&= \sum_{t=0} (c_t' + V_K(x_t') - V_K(x_t')) - V_K(x) \\
&= \sum_{t=0} (c_t' + V_K(x_{t+1}') - V_K(x_t')) \\
&= \sum_{t=0} A_K(x_t', u_t')
\end{aligned}
$$

which completes the first claim.

For the second claim, observe that:

$$
V_K(x) = x^\top \left( Q + K^\top R K \right) x + x^\top (A - BK)^\top P_K (A - BK) x
$$

And, for $u = K'x$,

$$
\begin{aligned}
A_K(x, u) &= Q_K(x, u) - V_K(x) \\
&= x^\top \left( Q + (K')^\top R K' \right) x + x^\top (A - BK')^\top P_K (A - BK') x - V_K(x) \\
&= x^\top \left( Q + (K' - K + K)^\top R (K' - K + K) \right) x + \\
&\quad x^\top (A - BK - B(K' - K))^\top P_K (A - BK - B(K' - K)) x - V_K(x) \\
&= 2x^\top (K' - K)^\top \left( (R + B^\top P_K B) K - B^\top P_K A \right) x + \\
&\quad x^\top (K' - K)^\top (R + B^\top P_K B)(K' - K)) x \,,
\end{aligned}
$$

which completes the proof. $\qquad\square$

This lemma is helpful in proving that $C(K)$ is gradient dominated.

**Lemma 8.** *(Gradient domination) Let $K^*$ be an optimal policy. Suppose $K$ has finite cost and $\mu > 0$. It holds that:*

$$
\begin{aligned}
C(K) - C(K^*) &\le \|\Sigma_{K^*}\| \mathrm{Tr}(E_K^\top (R + B^\top P_K B)^{-1} E_K) \\
&\le \frac{\|\Sigma_{K^*}\|}{\sigma_{min}(R)} \mathrm{Tr}(E_K^\top E_K) \\
&\le \frac{\|\Sigma_{K^*}\|}{\mu^2 \sigma_{min}(R)} \mathrm{Tr}(\nabla C(K)^\top \nabla C(K))
\end{aligned}
$$

*For a lower bound, it holds that:*

$$
C(K) - C(K^*) \ge \frac{\mu}{\|R + B^\top P_K B\|} \mathrm{Tr}(E_K^\top E_K)
$$

*Proof.* From Equation 10 and by completing the square,

$$
\begin{aligned}
& Q_K(x, K'x) - V_K(x) \\
=\ & 2\mathrm{Tr}(xx^\top (K' - K)^\top E_K) + \mathrm{Tr}(xx^\top (K' - K)^\top (R + B^\top P_K B)(K' - K)) \\
=\ & \mathrm{Tr}(xx^\top \left( K' - K + (R + B^\top P_K B)^{-1} E_K \right)^\top (R + B^\top P_K B) \left( K' - K + (R + B^\top P_K B)^{-1} E_K \right)) \\
& -\mathrm{Tr}(xx^\top E_K^\top (R + B^\top P_K B)^{-1} E_K) \\
\ge\ & -\mathrm{Tr}(xx^\top E_K^\top (R + B^\top P_K B)^{-1} E_K) \quad\quad (11)
\end{aligned}
$$

with equality when $K' = K - (R + B^\top P_K B)^{-1} E_K$.

Let $x_t^*$ and $u_t^*$ be the sequence generated under $K_*$. Using this and Lemma 7,

$$
\begin{aligned}
C(K) - C(K^*) \;&=\; -\mathbb{E}\sum_t A_K(x_t^*, u_t^*) \\
&\leq\; \mathbb{E}\sum_t \mathrm{Tr}(x_t^*(x_t^*)^\top E_K^\top (R + B^\top P_K B)^{-1} E_K) \\
&=\; \mathrm{Tr}(\Sigma_{K^*} E_K^\top (R + B^\top P_K B)^{-1} E_K) \\
&\leq\; \|\Sigma_{K^*}\|\mathrm{Tr}(E_K^\top (R + B^\top P_K B)^{-1} E_K) \\
&\leq\; \|\Sigma_{K^*}\|\|(R + B^\top P_K B)^{-1}\|\,\mathrm{Tr}(E_K^\top E_K) \\
&\leq\; \frac{\|\Sigma_{K^*}\|}{\sigma_{\min}(R)}\mathrm{Tr}(E_K^\top E_K) \\
&=\; \frac{\|\Sigma_{K^*}\|}{\sigma_{\min}(R)}\mathrm{Tr}(\Sigma_K^{-1}\nabla C(K)^\top \nabla C(K)\Sigma_K^{-1}) \\
&\leq\; \frac{\|\Sigma_{K^*}\|}{\mu^2\sigma_{\min}(R)}\mathrm{Tr}(\nabla C(K)^\top \nabla C(K))
\end{aligned}
$$

which completes the proof of the upper bound.

For the lower bound, consider $K' = K - (R + B^\top P_K B)^{-1}E_K$ where equality holds in Equation 11. Let $x_t'$ and $u_t'$ be the sequence generated under $K'$. Using that $C(K^*) \leq C(K')$,

$$
\begin{aligned}
C(K) - C(K^*) \;&\geq\; C(K) - C(K') \\
&=\; -\mathbb{E}\sum_t A_K(x_t', u_t') \\
&=\; \mathbb{E}\sum_t \mathrm{Tr}(x_t'(x_t')^\top E_K^\top (R + B^\top P_K B)^{-1} E_K) \\
&\geq\; \mathrm{Tr}(\Sigma_{K'} E_K^\top (R + B^\top P_K B)^{-1} E_K) \\
&\geq\; \frac{\mu}{\|R + B^\top P_K B\|}\mathrm{Tr}(E_K^\top E_K)
\end{aligned}
$$

which completes the proof. $\qquad\square$

Recall that a function $f$ is said to be smooth (or $C^1$-smooth) if it satisfies for some finite $\beta$, it satisfies:

$$
|f(x) - f(y) - \nabla f(y)^\top(x - y)| \leq \frac{\beta}{2}\|x - y\|^2\,. \tag{12}
$$

for all $x, y$ (equivalently, it is smooth if the gradients of $f$ are continuous).

**Lemma 9.** *("Almost" smoothness) $C(K)$ satisfies:*

$$
C(K') - C(K) = -2\mathrm{Tr}(\Sigma_{K'}(K - K')^\top E_K) + \mathrm{Tr}(\Sigma_{K'}(K - K')^\top(R + B^\top P_K B)(K - K'))
$$

To see why this is related to smoothness (e.g. compare to Equation 12), suppose $K'$ is sufficiently close to $K$ so that:

$$
\Sigma_{K'} \approx \Sigma_K + O(\|K - K'\|) \tag{13}
$$

and the leading order term $2\mathrm{Tr}(\Sigma_{K'}(K' - K)^\top E_K)$ would then behave as $\mathrm{Tr}((K' - K)^\top \nabla C(K))$. The challenge in the proof (for gradient descent) is quantifying the lower order terms in this argument.

*Proof.* The claim immediately results from Lemma 7, by using Equation 10 and taking an expectation. $\qquad\square$

The next lemma spectral norm bounds on $P_K$ and $\Sigma_K$ are helpful:

**Lemma 10.** *It holds that:*

$$
\|P_K\| \leq \frac{C(K)}{\mu}, \qquad \|\Sigma_K\| \leq \frac{C(K)}{\sigma_{\min}(Q)}
$$

*Proof.* For the first claim, $C(K)$ is lower bounded as:

$$C(K) = \mathbb{E}_{x_0 \sim \mathcal{D}} x_0^\top P_K x_0 \geq \|P_K\| \sigma_{\min}(\mathbb{E} x_0 x_0^\top)$$

Alternatively, $C(K)$ can be lower bounded as:

$$C(K) = \text{Tr}(\Sigma_K(Q + K^\top RK)) \geq \text{Tr}(\Sigma_K)\sigma_{\min}(Q) \geq \|\Sigma_K\|\sigma_{\min}(Q) \,,$$

which proves the second claim. □

### C.2 GAUSS-NEWTON ANALYSIS

The next lemma bounds the one step progress of Gauss-Newton.

**Lemma 11.** *Suppose that:*

$$K' = K - \eta(R + B^\top P_K B)^{-1} \nabla C(K) \Sigma_K^{-1} \,, .$$

*If $\eta \leq 1$, then*

$$C(K') - C(K^*) \leq \left(1 - \frac{\eta\mu}{\|\Sigma_{K^*}\|}\right)(C(K) - C(K^*))$$

*Proof.* Observe $K' = K - \eta(R + B^\top P_K B)^{-1} E_K$. Using Lemma 9 and the condition on $\eta$,

$$
\begin{aligned}
C(K') - C(K) &= -2\eta\text{Tr}(\Sigma_{K'} E_K^\top (R + B^\top P_K B)^{-1} E_K) + \eta^2\text{Tr}(\Sigma_{K'} E_K^\top (R + B^\top P_K B)^{-1} E_K) \\
&\leq -\eta\text{Tr}(\Sigma_{K'} E_K^\top (R + B^\top P_K B)^{-1} E_K) \\
&\leq -\eta\sigma_{\min}(\Sigma_{K'})\text{Tr}(E_K^\top (R + B^\top P_K B)^{-1} E_K) \\
&\leq -\eta\mu\text{Tr}(E_K^\top (R + B^\top P_K B)^{-1} E_K) \\
&\leq -\eta\frac{\mu}{\|\Sigma_{K^*}\|}(C(K) - C(K^*)) \,,
\end{aligned}
$$

where the last step uses Lemma 8. □

With this lemma, the proof of the convergence rate of the Gauss Newton algorithm is immediate.

*Proof.* (of Theorem 5, Gauss-Newton case) The theorem is due to that $\eta = 1$ leads to a contraction of $1 - \frac{\eta\mu}{\|\Sigma_{K^*}\|}$ at every step. □

### C.3 NATURAL POLICY GRADIENT DESCENT ANALYSIS

The next lemma bounds the one step progress of the natural policy gradient.

**Lemma 12.** *Suppose:*

$$K' = K - \eta\nabla C(K)\Sigma_K^{-1}$$

*and that $\eta \leq \frac{1}{\|R + B^\top P_K B\|}$. It holds that:*

$$C(K') - C(K^*) \leq \left(1 - \eta\sigma_{min}(R)\frac{\mu}{\|\Sigma_{K^*}\|}\right)(C(K) - C(K^*))$$

*Proof.* Since $K' = K - \eta E_K$, Lemma 9 implies:

$$C(K') - C(K) = -2\eta\text{Tr}(\Sigma_{K'} E_K^\top E_K) + \eta^2\text{Tr}(\Sigma_{K'} E_K^\top (R + B^\top P_K B) E_K)$$

The last term can be bounded as:

$$
\begin{aligned}
\text{Tr}(\Sigma_{K'} E_K^\top (R + B^\top P_K B) E_K) &= \text{Tr}((R + B^\top P_K B) E_K \Sigma_{K'} E_K^\top) \\
&\leq \|R + B^\top P_K B\|\text{Tr}(E_K \Sigma_{K'} E_K^\top) \\
&= \|R + B^\top P_K B\|\text{Tr}(\Sigma_{K'} E_K^\top E_K) \,.
\end{aligned}
$$

Continuing and using the condition on $\eta$,

$$
\begin{aligned}
C(K') - C(K) &\leq -2\eta \mathrm{Tr}(\Sigma_{K'} E_K^\top E_K) + \eta^2 \|R + B^\top P_K B\| \mathrm{Tr}(\Sigma_{K'} E_K^\top E_K) \\
&\leq -\eta \mathrm{Tr}(\Sigma_{K'} E_K^\top E_K) \\
&\leq -\eta \sigma_{\min}(\Sigma_{K'}) \mathrm{Tr}(E_K^\top E_K) \\
&\leq -\eta \mu \mathrm{Tr}(E_K^\top E_K) \\
&\leq -\eta \frac{\mu \sigma_{\min}(R)}{\|\Sigma_{K^*}\|} (C(K) - C(K^*))
\end{aligned}
$$

using Lemma 8. $\qquad\square$

With this lemma, the proof of the natural policy gradient convergence rate can be completed.

*Proof.* (of Theorem 5, natural policy gradient case) Using Lemma 10,

$$
\frac{1}{\|R + B^\top P_K B\|} \geq \frac{1}{\|R\| + \|B\|^2 \|P_K\|} \geq \frac{1}{\|R\| + \frac{\|B\|^2 C(K)}{\mu}}
$$

The proof is completed by induction: $C(K_1) \leq C(K_0)$, since Lemma 12 can be applied. The proof proceeds by arguing that Lemma 12 can be applied at every step. If it were the case that $C(K_t) \leq C(K_0)$, then

$$
\eta \leq \frac{1}{\|R\| + \frac{\|B\|^2 C(K_0)}{\mu}} \leq \frac{1}{\|R\| + \frac{\|B\|^2 C(K_t)}{\mu}} \leq \frac{1}{\|R + B^\top P_{K_t} B\|}
$$

and by Lemma 12:

$$
C(K_{t+1}) - C(K^*) \leq \left(1 - \frac{\mu}{\|\Sigma_{K^*}\|} \frac{\sigma_{\min}(R)}{\|R\| + \frac{\|B\|^2 C(K_0)}{\mu})}\right) (C(K_t) - C(K^*))
$$

which completes the proof. $\qquad\square$

## C.4 GRADIENT DESCENT ANALYSIS

As informally argued by Equation 13, the proof seeks to quantify how $\Sigma_{K'}$ changes with $\eta$. Then the proof bounds the one step progress of gradient descent.

### $\Sigma_K$ PERTURBATION ANALYSIS

This subsections aims to prove the following:

**Lemma 13.** *($\Sigma_K$ perturbation) Suppose $K'$ is such that:*

$$
\|K' - K\| \leq \frac{\sigma_{\min}(Q)\mu}{4C(K)\|B\| (\|A - BK\| + 1)}
$$

*It holds that:*

$$
\|\Sigma_{K'} - \Sigma_K\| \leq 4 \left(\frac{C(K)}{\sigma_{\min}(Q)}\right)^2 \frac{\|B\| (\|A - BK\| + 1)}{\mu} \|K - K'\|
$$

The proof proceeds by starting with a few technical lemmas. First, define a linear operator on symmetric matrices, $\mathcal{T}_K(\cdot)$, which can be viewed as a matrix on $\binom{d+1}{2}$ dimensions. Define this operator on a symmetric matrix $X$ as follows:

$$
\mathcal{T}_K(X) := \sum_{t=0}^\infty (A - BK)^t X [(A - BK)^\top]^t
$$

Also define the induced norm of $\mathcal{T}$ as follows:

$$
\|\mathcal{T}_K\| = \sup_X \frac{\|\mathcal{T}_K(X)\|}{\|X\|} \tag{14}
$$

where the supremum is over all symmetric matrices $X$ (whose spectral norm is non-zero).

Also, define

$$\Sigma_0 = \mathbb{E}x_0 x_0^\top$$

.

**Lemma 14.** *($\mathcal{T}_K$ norm bound) It holds that*

$$\|\mathcal{T}_K\| \leq \frac{C(K)}{\mu\,\sigma_{\min}(Q)}$$

*Proof.* For a unit norm vector $v \in \mathbb{R}^d$ and unit spectral norm matrix $X$,

$$
\begin{aligned}
v^\top(\mathcal{T}_K(X))v &= \sum_{t=0}^\infty v^\top(A-BK)^t X[(A-BK)^\top]^t v \\
&= \sum_{t=0}^\infty \mathrm{Tr}([(A-BK)^\top]^t vv^\top(A-BK)^t X) \\
&= \sum_{t=0}^\infty \mathrm{Tr}([\Sigma_0^{1/2}(A-BK)^\top]^t vv^\top(A-BK)^t \Sigma_0^{1/2}\Sigma_0^{-1/2}X\Sigma_0^{-1/2}) \\
&\leq \sum_{t=0}^\infty \mathrm{Tr}([\Sigma_0^{1/2}(A-BK)^\top]^t vv^\top(A-BK)^t \Sigma_0^{1/2})\|\Sigma_0^{-1/2}X\Sigma_0^{-1/2}\| \\
&= \|\Sigma_0^{-1/2}X\Sigma_0^{-1/2}\|\ \left(v^\top\mathcal{T}_K(\Sigma_0)v\right) \\
&\leq \frac{1}{\sigma_{\min}(\mathbb{E}x_0 x_0^\top)}\|\mathcal{T}_K(\Sigma_0)\| \\
&= \frac{1}{\mu}\|\Sigma_K\|
\end{aligned}
$$

using that $\mathcal{T}_K(\Sigma_0) = \Sigma_K$. The proof is completed using the upper bound on $\|\Sigma_K\|$ in Lemma 10. $\square$

Also, with respect to $K$, define another linear operator on symmetric matrices:

$$\mathcal{F}_K(X) = (A-BK)X(A-BK)^\top.$$

Let I to denote the identity operator on the same space. Define the induced norm $\|\cdot\|$ of these operators as in Equation 14. Note these operators are related to the operator $\mathcal{T}_K$ as follows:

**Lemma 15.** *When $(A-BK)$ has spectral radius smaller than 1,*

$$\mathcal{T}_K = (\mathrm{I} - \mathcal{F}_K)^{-1}.$$

*Proof.* When $(A-BK)$ has spectral radius smaller than 1, $\mathcal{T}_K$ is well defined and is the solution of $\mathcal{T}_K = \mathrm{I} + \mathcal{T}_K \circ \mathcal{F}_K$. Therefore $\mathcal{T}_K \circ (\mathrm{I} - \mathcal{F}_K) = \mathrm{I}$ and $\mathcal{T}_K = (\mathrm{I} - \mathcal{F}_K)^{-1}$. $\square$

Since,

$$\Sigma_K = \mathcal{T}_K(\Sigma_0) = (\mathrm{I} - \mathcal{F}_K)^{-1}(\Sigma_0).$$

The proof of Lemma 13 seeks to bound:

$$\|\Sigma_K - \Sigma_{K'}\| = \|(\mathcal{T}_K - \mathcal{T}_{K'})(\Sigma_0)\| = \|((\mathrm{I}-\mathcal{F}_K)^{-1} - (\mathrm{I}-\mathcal{F}_{K'})^{-1})(\Sigma_0)\|.$$

The following two perturbation bounds are helpful in this.

**Lemma 16.** *It holds that:*

$$\|\mathcal{F}_K - \mathcal{F}_{K'}\| \leq 2\|A-BK\|\|B\|\|K-K'\| + \|B\|^2\|K-K'\|^2.$$

*Proof.* Let $\Delta = K - K'$. For every matrix $X$,

$$(\mathcal{F}_K - \mathcal{F}_{K'})(X) = (A - BK)X(B\Delta)^\top + (B\Delta)X(A - BK)^\top - (B\Delta)X(B\Delta)^\top.$$

The operator norm of $\mathcal{F}_K - \mathcal{F}_{K'}$ is the maximum possible ratio in spectral norm of $(\mathcal{F}_K - \mathcal{F}_{K'})(X)$ and $X$. Then the claim follows because $\|AX\| \le \|A\|\|X\|$. □

**Lemma 17.** *If*

$$\|\mathcal{T}_K\|\|\mathcal{F}_K - \mathcal{F}_{K'}\| \le 1/2\,,$$

*then*

$$\begin{aligned} \|\left(\mathcal{T}_K - \mathcal{T}_{K'}\right)(\Sigma)\| &\le 2\|\mathcal{T}_K\|\|\mathcal{F}_K - \mathcal{F}_{K'}\|\|\mathcal{T}_K(\Sigma)\|. \\ &\le 2\|\mathcal{T}_K\|^2\|\mathcal{F}_K - \mathcal{F}_{K'}\|\|\Sigma\|. \end{aligned}$$

*Proof.* Define $\mathcal{A} = I - \mathcal{F}_K$, and $\mathcal{B} = \mathcal{F}_{K'} - \mathcal{F}_K$. In this case $\mathcal{A}^{-1} = \mathcal{T}_K$ and $(\mathcal{A} - \mathcal{B})^{-1} = \mathcal{T}_{K'}$. Hence, the condition $\|\mathcal{T}_K\|\|\mathcal{F}_K - \mathcal{F}_{K'}\| \le 1/2$ translates to the condition $\|\mathcal{A}^{-1}\|\|\mathcal{B}\| \le 1/2$.

Observe:

$$(\mathcal{A}^{-1} - (\mathcal{A} - \mathcal{B})^{-1})(\Sigma) = (I - (I - \mathcal{A}^{-1} \circ \mathcal{B})^{-1})(\mathcal{A}^{-1}(\Sigma)) = (I - (I - \mathcal{A}^{-1} \circ \mathcal{B})^{-1})(\mathcal{T}_K(\Sigma))\,.$$

Since $(I - \mathcal{A}^{-1} \circ \mathcal{B})^{-1} = I + \mathcal{A}^{-1} \circ \mathcal{B} \circ (I - \mathcal{A}^{-1} \circ \mathcal{B})^{-1}$,

$$\|(I - \mathcal{A}^{-1} \circ \mathcal{B})^{-1}\| \le 1 + \|\mathcal{A}^{-1} \circ \mathcal{B}\|\|(I - \mathcal{A}^{-1} \circ \mathcal{B})^{-1}\| \le 1 + 1/2\|(I - \mathcal{A}^{-1} \circ \mathcal{B})^{-1}\|$$

which implies $\|(I - \mathcal{A}^{-1} \circ \mathcal{B})^{-1}\| \le 2$. Hence,

$$\|I - (I - \mathcal{A}^{-1} \circ \mathcal{B})^{-1}\| = \|\mathcal{A}^{-1} \circ \mathcal{B} \circ (I - \mathcal{A}^{-1} \circ \mathcal{B})^{-1}\| \le \|\mathcal{A}^{-1}\|\|\mathcal{B}\|\|(I - \mathcal{A}^{-1} \circ \mathcal{B})^{-1}\| = 2\|\mathcal{A}^{-1}\|\|\mathcal{B}\|.$$

and so

$$\|I - (I - \mathcal{A}^{-1} \circ \mathcal{B})^{-1}\| \le 2\|\mathcal{A}^{-1}\|\|\mathcal{B}\| = 2\|\mathcal{T}_K\|\|\mathcal{F}_K - \mathcal{F}_{K'}\|\,.$$

Combining these two,

$$\|\left(\mathcal{T}_K - \mathcal{T}_{K'}\right)(\Sigma)\| \le \|(I - (I - \mathcal{A}^{-1} \circ \mathcal{B})^{-1})\|\|\mathcal{T}_K(\Sigma)\| \le 2\|\mathcal{T}_K\|\|\mathcal{F}_K - \mathcal{F}_{K'}\|\|\mathcal{T}_K(\Sigma)\|.$$

This proves the main inequality. The last step of the inequality is just applying definition of the norm of $\mathcal{T}_K$: $\|\mathcal{T}_K(\Sigma)\| \le \|\mathcal{T}_K\|\|\Sigma\|$. □

With these Lemmas, the proof is completed as follows:

*Proof.* (of Lemma 13) First, the proof shows $\|\mathcal{T}_K\|\|\mathcal{F}_K - \mathcal{F}_{K'}\| \le 1/2$, which is the desired condition in Lemma 17. First, observe that under the assumed condition on $\|K - K'\|$, implies that

$$\|B\|\|K' - K\| \le \frac{\sigma_{\min}(Q)\mu}{4C(K)\left(\|A - BK\| + 1\right)} \le \frac{1}{4}\frac{\sigma_{\min}(Q)\mu}{C(K)} \le \frac{1}{4}$$

using that $\frac{\sigma_{\min}(Q)\mu}{C(K)} \le 1$ due to Lemma 10. Using Lemma 16,

$$\begin{aligned} \|\mathcal{F}_K - \mathcal{F}_{K'}\| &\le \left(2\|A - BK\|\|B\|\|K - K'\| + \|B\|^2\|K - K'\|^2\right) \\ &\le 2\|B\|\left(\|A - BK\| + 1\right)\|K - K'\| \end{aligned} \tag{15}$$

Using this and Lemma 14,

$$\|\mathcal{T}_K\|\|\mathcal{F}_K - \mathcal{F}_{K'}\| \le \frac{C(K)}{\sigma_{\min}(Q)\mu}2\|B\|\left(\|A - BK\| + 1\right)\|K - K'\| \le \frac{1}{2}$$

where the last step uses the condition on $\|K - K'\|$.

Thus,

$$\begin{aligned} \|\Sigma_{K'} - \Sigma_K\| &\le 2\|\mathcal{T}_K\|\|\mathcal{F}_K - \mathcal{F}_{K'}\|\|\mathcal{T}_K(\Sigma_0)\| \\ &\le 2\frac{C(K)}{\sigma_{\min}(Q)\mu}\left(2\|B\|\left(\|A - BK\| + 1\right)\|K - K'\|\right)\frac{C(K)}{\sigma_{\min}(Q)} \end{aligned}$$

using Lemmas 10 and 16. □

GRADIENT DESCENT PROGRESS

Equipped with these lemmas, the one step progress of gradient descent can be bounded.

**Lemma 18.** *Suppose that*

$$K' = K - \eta \nabla C(K),$$

*where*

$$\eta \leq \frac{1}{16} \min \left\{ \left( \frac{\sigma_{\min}(Q)\mu}{C(K)} \right)^2 \frac{1}{\|B\| \|\nabla C(K)\| (1 + \|A - BK\|)}, \frac{\sigma_{\min}(Q)}{2C(K)\|R + B^\top P_K B\|} \right\}.$$

(16)

*It holds that:*

$$C(K') - C(K^*) \leq \left( 1 - \eta \sigma_{min}(R) \frac{\mu^2}{\|\Sigma_{K^*}\|} \right) (C(K) - C(K^*))$$

*Proof.* By Lemma 9,

$$
\begin{aligned}
& C(K') - C(K) \\
=\ & -2\eta \mathrm{Tr}(\Sigma_{K'} \Sigma_K E_K^\top E_K) + \eta^2 \mathrm{Tr}(\Sigma_K \Sigma_{K'} \Sigma_K E_K^\top (R + B^\top P_K B) E_K) \\
\leq\ & -2\eta \mathrm{Tr}(\Sigma_K E_K^\top E_K \Sigma_K) + 2\eta \|\Sigma_{K'} - \Sigma_K\| \mathrm{Tr}(\Sigma_K E_K^\top E_K) \\
& + \eta^2 \|\Sigma_{K'}\| \|R + B^\top P_K B\| \mathrm{Tr}(\Sigma_K \Sigma_K E_K^\top E_K) \\
\leq\ & -2\eta \mathrm{Tr}(\Sigma_K E_K^\top E_K \Sigma_K) + 2\eta \frac{\|\Sigma_{K'} - \Sigma_K\|}{\sigma_{\min}(\Sigma_K)} \mathrm{Tr}(\Sigma_K E_K^\top E_K \Sigma_K) \\
& + \eta^2 \|\Sigma_{K'}\| \|R + B^\top P_K B\| \mathrm{Tr}(\Sigma_K E_K^\top E_K \Sigma_K) \\
=\ & -2\eta \left( 1 - \frac{\|\Sigma_{K'} - \Sigma_K\|}{\sigma_{\min}(\Sigma_K)} - \frac{\eta}{2} \|\Sigma_{K'}\| \|R + B^\top P_K B\| \right) \mathrm{Tr}(\nabla C(K)^\top \nabla C(K)) \\
\leq\ & -2\eta \frac{\mu^2 \sigma_{\min}(R)}{\|\Sigma_{K^*}\|} \left( 1 - \frac{\|\Sigma_{K'} - \Sigma_K\|}{\mu} - \frac{\eta}{2} \|\Sigma_{K'}\| \|R + B^\top P_K B\| \right) (C(K) - C(K^*))
\end{aligned}
$$

where the last step uses Lemma 8.

By Lemma 13,

$$\frac{\|\Sigma_{K'} - \Sigma_K\|}{\mu} \leq 4\eta \left( \frac{C(K)}{\sigma_{\min}(Q)\mu} \right)^2 \|B\| (\|A - BK\| + 1)) \|\nabla C(K)\| \leq 1/4$$

using the assumed condition on $\eta$.

Using this last claim and Lemma 10,

$$\|\Sigma_{K'}\| \leq \|\Sigma_{K'} - \Sigma_K\| + \|\Sigma_K\| \leq \frac{\mu}{4} + \frac{C(K)}{\sigma_{\min}(Q)} \leq \frac{\|\Sigma_{K'}\|}{4} + \frac{C(K)}{\sigma_{\min}(Q)}$$

and so $\|\Sigma_{K'}\| \leq \frac{4C(K)}{3\sigma_{\min}(Q)}$. Hence,

$$1 - \frac{\|\Sigma_{K'} - \Sigma_K\|}{\mu} - \frac{\eta}{2} \|\Sigma_{K'}\| \|R + B^\top P_K B\| \geq 1 - 1/4 - \frac{\eta}{2} \frac{4C(K)}{3\sigma_{\min}(Q)} \|R + B^\top P_K B\| \geq 1/2$$

using the condition on $\eta$. □

In order to prove a gradient descent convergence rate, the following bounds are helpful:

**Lemma 19.** *It holds that*

$$\|\nabla C(K)\| \leq \frac{C(K)}{\sigma_{\min}(Q)} \sqrt{\frac{\|R + B^\top P_K B\| (C(K) - C(K^*))}{\mu}}$$

*and that:*

$$\|K\| \leq \frac{1}{\sigma_{\min}(R)} \left( \sqrt{\frac{\|R + B^\top P_K B\| (C(K) - C(K^*))}{\mu}} + \|B^\top P_K A\| \right)$$

*Proof.* Using Lemma 10,

$$\|\nabla C(K)\|^2 \leq \mathrm{Tr}(\Sigma_K E_K^\top E_K \Sigma_K) \leq \|\Sigma_K\|^2 \mathrm{Tr}(E_K^\top E_K) \leq \left(\frac{C(K)}{\sigma_{\min}(Q)}\right)^2 \mathrm{Tr}(E_K^\top E_K)$$

By Lemma 8,

$$\mathrm{Tr}(E_K^\top E_K) \leq \frac{\|R + B^\top P_K B\|(C(K) - C(K^*))}{\mu}$$

which proves the first claim.

Again using Lemma 8,

$$\begin{aligned}
\|K\| &\leq& \|(R + B^\top P_K B)^{-1}\| \|(R + B^\top P_K B)K\| \\
&\leq& \frac{1}{\sigma_{\min}(R)}\|(R + B^\top P_K B)K\| \\
&\leq& \frac{1}{\sigma_{\min}(R)}\left(\|(R + B^\top P_K B)K - B^\top P_K A\| + \|B^\top P_K A\|\right) \\
&=& \frac{\|E_K\|}{\sigma_{\min}(R)} + \frac{\|B^\top P_K A\|}{\sigma_{\min}(R)} \\
&\leq& \frac{\sqrt{\mathrm{Tr}(E_K^\top E_K)}}{\sigma_{\min}(R)} + \frac{\|B^\top P_K A\|}{\sigma_{\min}(R)} \\
&=& \frac{\sqrt{(C(K) - C(K^*))\|R + B^\top P_K B\|}}{\sqrt{\mu}\sigma_{\min}(R)} + \frac{\|B^\top P_K A\|}{\sigma_{\min}(R)}
\end{aligned}$$

which proves the second claim. □

With these lemmas, the proof of the gradient descent convergence rate follows:

*Proof.* (of Theorem 5, gradient descent case) First, the following argues that progress is made at $t = 1$. Based on Lemma 10 and Lemma 19, by choosing $\eta$ to be an appropriate polynomial in $\frac{1}{C(K_0)}, \frac{1}{\|A\|}, \frac{1}{\|B\|}, \frac{1}{\|R\|}, \sigma_{\min}(R), \sigma_{\min}(Q)$ and $\mu$, the stepsize condition in Equation 16 is satisfied. Hence, by Lemma 18,

$$C(K_1) - C(K^*) \leq \left(1 - \eta\sigma_{\min}(R)\frac{\mu^2}{\|\Sigma_{K^*}\|}\right)(C(K_0) - C(K^*))$$

which implies that the cost decreases at $t = 1$. Proceeding inductively, now suppose that $C(K_t) \leq C(K_0)$, then the stepsize condition in Equation 16 is still satisfied (due to the use of $C(K_0)$ in bounding the quantities in Lemma 19). Thus, Lemma 18 can again be applied for the update at time $t + 1$ to obtain:

$$C(K_{t+1}) - C(K^*) \leq \left(1 - \eta\sigma_{\min}(R)\frac{\mu^2}{\|\Sigma_{K^*}\|}\right)(C(K_t) - C(K^*)).$$

Provided

$$T \geq \frac{\|\Sigma_{K^*}\|}{\eta\mu^2\sigma_{\min}(R)}\log\frac{C(K_0) - C(K^*)}{\varepsilon},$$

then $C(K_T) - C(K^*) \leq \varepsilon$, and the result follows. □

## D   ANALYSIS: THE MODEL-FREE CASE

This section shows how techniques from zeroth order optimization allow the algorithm to run in the model-free setting with only black-box access to a simulator. The dependencies on various parameters are not optimized, and the notation $h$ is used to represent different polynomial factors in the relevant factors ($\frac{C(K_0)}{\mu\sigma_{min}(Q)}, \|A\|, \|B\|, \|R\|, 1/\sigma_{min}(R)$). When the polynomial also depend on dimension $d$ or accuracy $1/\epsilon$, this is specified as parameters ($h(d, 1/\epsilon)$).

The section starts by showing how the infinite horizon can be approximated with a finite horizon.

### D.1 Approximating $C(K)$ and $\Sigma_K$ with finite horizon

This section shows that as long as there is an upper bound on $C(K)$, it is possible to approximate both $C(K)$ and $\Sigma(K)$ with any desired accuracy.

**Lemma 20.** *For any $K$ with finite $C(K)$, let $\Sigma_K^{(\ell)} = \mathbb{E}[\sum_{i=0}^{\ell-1} x_i x_i^\top]$ and $C^{(\ell)}(K) = \mathbb{E}[\sum_{i=0}^{\ell-1} x_i^\top Q x_i + u_i^\top R u_i] = \langle \Sigma_K^{(t)}, Q + K^\top R K \rangle$. If*

$$\ell \geq \frac{d \cdot C^2(K)}{\epsilon \mu \sigma_{min}^2(Q)},$$

*then $\|\Sigma_K^{(\ell)} - \Sigma_K\| \leq \epsilon$. Also, if*

$$\ell \geq \frac{d \cdot C^2(K)(\|Q\| + \|R\|\|K\|^2)}{\epsilon \mu \sigma_{min}^2(Q)}$$

*then $C(K) \geq C^{(\ell)}(K) \geq C(K) - \epsilon$.*

*Proof.* First, the bound on $\Sigma_K$ is proved. Define the operators $\mathcal{T}_K$ and $\mathcal{F}_K$ as in Section C.4, observe $\Sigma_K = \mathcal{T}_K(\Sigma_0)$ and $\Sigma_K^{(\ell)} = \Sigma_K - (\mathcal{F}_K)^\ell(\Sigma_K)$.

If $X \succeq Y$, then $\mathcal{F}_K(X) \succeq \mathcal{F}_K(Y)$, this follows immediately from the form of $\mathcal{F}_K(X) = (A + BK)X(A + BK)^\top$. If $X$ is PSD then $WXW^\top$ is also PSD for any $W$.

Now, since

$$\text{tr}(\sum_{i=0}^{\ell-1} \mathcal{F}^\ell(\Sigma_0)) \leq \text{tr}(\sum_{i=0}^{\infty} \mathcal{F}^\ell(\Sigma_0)) = \text{tr}(\Sigma_K) \leq \frac{d \cdot C(K)}{\sigma_{min}(Q)}.$$

(Here the last step is by Lemma 10), and all traces are nonnegative, then there must exists $j < \ell$ such that $\text{tr}(\mathcal{F}_K^j(\Sigma_0)) \leq \frac{d \cdot C(K)}{\ell \sigma_{min}(Q)}$.

Also, since $\Sigma_K \preceq \frac{C(K)}{\mu \sigma_{min}(Q)} \Sigma_0$,

$$\text{tr}(\mathcal{F}_K^j(\Sigma_K)) \leq \frac{C(K)}{\mu \sigma_{min}(Q)} \text{tr}(\mathcal{F}_K^j(\Sigma_0)) \leq \frac{d \cdot C^2(K)}{t \mu \sigma_{min}^2(Q)}.$$

Therefore as long as

$$\ell \geq \frac{dC^2(K)}{\epsilon \mu \sigma_{min}^2(Q)},$$

it follows that:

$$\|\Sigma_K - \Sigma_K^{(\ell)}\| \leq \|\Sigma_K - \Sigma_K^{(j)}\| = \|\mathcal{F}^j(\Sigma_K)\| \leq \epsilon.$$

Here the first step is again because of all the terms are PSD, so using more terms is always better. The last step follows because $\mathcal{F}^j(\Sigma_K)$ is also a PSD matrix so the spectral norm is bounded by trace. In fact, it holds that $\text{tr}(\Sigma_K - \Sigma_K^{(\ell)})$ is smaller than $\epsilon$.

Next, observe $C^{(\ell)}(K) = \langle \Sigma_K^{(\ell)}, Q + K^\top R K \rangle$ and $C(K) = \langle \Sigma_K, Q + K^\top R K \rangle$, therefore

$$C(K) - C^{(\ell)}(K) \leq \text{tr}(\Sigma_K - \Sigma_K^{(\ell)})(\|Q\| + \|R\|\|K\|^2).$$

Therefore if

$$\ell \geq \frac{d \cdot C^2(K)(\|Q\| + \|R\|\|K\|^2)}{\epsilon \mu \sigma_{min}^2(Q)},$$

then $\text{tr}(\Sigma_K - \Sigma_K^{(\ell)}) \leq \epsilon/(\|Q\| + \|R\|\|K\|^2)$ and hence $C(K) - C^{(\ell)}(K) \leq \epsilon$.

$\square$

## D.2   Perturbation of $C(K)$ and $\nabla C(K)$

The next lemma show that the function value and its gradient are approximate preserved if a small perturbation to the policy $K$ is applied.

**Lemma 21.** *($C_K$ perturbation) Suppose $K'$ is such that:*

$$\|K' - K\| \leq \min\left(\frac{\sigma_{\min}(Q)\mu}{4C(K)\|B\|\,(\|A - BK\| + 1)}, \|K\|\right)$$

*then:*

$$|C(K') - C(K)|$$
$$\leq \quad 6\|K\|\,\|R\|\,\mathbb{E}\|x_0\|^2\left(\frac{C(K)}{\mu\,\sigma_{\min}(Q)}\right)^2 (\|K\|\|B\|\|A - BK\| + \|K\|\|B\| + 1)\,\|K - K'\|$$

*Proof.* As in the proof of Lemma 16, the assumption implies that $\|\mathcal{T}_K\|\|\mathcal{F}_K - \mathcal{F}_{K'}\| \leq 1/2$, and, from Equation 15, that

$$\|\mathcal{F}_K - \mathcal{F}_{K'}\| \leq 2\|B\|\,(\|A - BK\| + 1)\,\|K - K'\|$$

First, observe:

$$
\begin{aligned}
C(K') - C(K) &\leq \quad \mathrm{Tr}(\mathbb{E}x_0 x_0^\top)\|\mathcal{T}_{K'}(Q + (K')^\top RK') - \mathcal{T}_K(Q + K^\top RK)\| \\
&= \quad \mathbb{E}\|x_0\|^2\|\mathcal{T}_{K'}(Q + (K')^\top RK') - \mathcal{T}_K(Q + K^\top RK)\|
\end{aligned}
$$

Hence,

$$
\begin{aligned}
&\|\mathcal{T}_{K'}(Q + (K')^\top RK') - \mathcal{T}_K(Q + K^\top RK)\| \\
\leq \quad &\|\mathcal{T}_{K'}(Q + (K')^\top RK') - \mathcal{T}_K(Q + (K')^\top RK') \\
&- \left(\mathcal{T}_K(Q + K^\top RK) - \mathcal{T}_K(Q + (K')^\top RK')\right)\| \\
= \quad &\|\mathcal{T}_{K'}(Q + (K')^\top RK') - \mathcal{T}_K(Q + (K')^\top RK') - \mathcal{T}_K \circ (K^\top RK - (K')^\top RK')\| \\
\leq \quad &2\|\mathcal{T}_K\|^2\|\mathcal{F}_K - \mathcal{F}_{K'}\|\|(K')^\top RK')\| + \|\mathcal{T}_K\|\|K^\top RK - (K')^\top RK'\| \\
\leq \quad &2\|\mathcal{T}_K\|^2\|\mathcal{F}_K - \mathcal{F}_{K'}\|\left(\|(K')^\top RK') - K^\top RK\| + \|K^\top RK)\|\right) \\
&+\|\mathcal{T}_K\|\|K^\top RK - (K')^\top RK'\| \\
\leq \quad &\|\mathcal{T}_K\|\|(K')^\top RK') - K^\top RK\| + 2\|\mathcal{T}_K\|^2\|\mathcal{F}_K - \mathcal{F}_{K'}\|\|K^\top RK\| \\
&+\|\mathcal{T}_K\|\|K^\top RK - (K')^\top RK'\| \\
= \quad &2\|\mathcal{T}_K\|\|(K')^\top RK') - K^\top RK\| + 2\|\mathcal{T}_K\|^2\|\mathcal{F}_K - \mathcal{F}_{K'}\|\|K^\top RK\|
\end{aligned}
$$

For the first term,

$$
\begin{aligned}
2\|\mathcal{T}_K\|\|(K')^\top RK') - K^\top RK\| &\leq 2\|\mathcal{T}_K\|\left(2\|K\|\|R\|\|K' - K\| + \|R\|\|K' - K\|^2\right) \\
&\leq 2\|\mathcal{T}_K\|\left(3\|K\|\|R\|\|K' - K\|\right)
\end{aligned}
$$

using the assumption that $\|K' - K\| \leq \|K\|$. For the second term,

$$2\|\mathcal{T}_K\|^2\|\mathcal{F}_K - \mathcal{F}_{K'}\|\|K^\top RK\| \leq 2\|\mathcal{T}_K\|^2\,2\|B\|\,(\|A - BK\| + 1)\,\|K - K'\|\,\|K\|^2\|R\|\,.$$

Combining the two terms completes the proof.   □

The next lemma shows the gradient is also stable after perturbation.

**Lemma 22.** *($\nabla C_K$ perturbation) Suppose $K'$ is such that:*

$$\|K' - K\| \leq \min\left(\frac{\sigma_{\min}(Q)\mu}{4C(K)\|B\|\,(\|A - BK\| + 1)}, \|K\|\right)$$

*then there is a polynomial $h_{grad}$ in $\frac{C(K_0)}{\mu\sigma_{min}(Q)}, \|A\|, \|B\|, \|R\|, \frac{1}{\sigma_{min}(R)}$ such that*

$$|\nabla C(K') - \nabla C(K)| \leq h_{grad}\|K' - K\|.$$

*Proof.* Recall $\nabla C(K) = 2E_K\Sigma_K$ where $E_K = (R + B^\top P_K B)K - B^\top P_K A$. Therefore

$$\nabla C(K') - \nabla C(K) = 2E_{K'}\Sigma_{K'} - 2E_K\Sigma_K = 2(E_{K'} - E_K)\Sigma_{K'} + 2E_K(\Sigma_{K'} - \Sigma_K).$$

Let's first look at the second term. By Lemma 8,

$$\mathrm{Tr}(E_K^\top E_K) \le \frac{\|R + B^\top P_K B\|(C(K) - C(K^*))}{\mu},$$

then by Lemma 13

$$\|\Sigma_{K'} - \Sigma_K\| \le 4\left(\frac{C(K)}{\sigma_{\min}(Q)}\right)^2 \frac{\|B\|(\|A - BK\| + 1)}{\mu}\|K - K'\|$$

Therefore the second term is bounded by

$$8\left(\frac{C(K)}{\sigma_{\min}(Q)}\right)^2 \frac{(\|R + B^\top P_K B\|(C(K) - C(K^*)))\|B\|(\|A - BK\| + 1)}{\mu^2}\|K - K'\|.$$

Now the first term is bounded. Since $K' - K$ is small enough, $\|\Sigma_{K'}\| \le \|\Sigma_K\| + \frac{C(K)}{\sigma_{min}(Q)}$.

For $E_{K'} - E_K$, a bound on $P_{K'} - P_K$ is provided. By the previous lemma,

$$\|P'_K - P_K\| = 6\left(\left(\frac{C(K)}{\mu\,\sigma_{\min}(Q)}\right)^2 \|K\|^2\|R\|\|B\|(\|A - BK\| + 1) + \left(\frac{C(K)}{\mu\,\sigma_{\min}(Q)}\right)\|K\|\|R\|\right)\|K - K'\|.$$

Therefore

$$E'_K - E_K = R(K' - K) + B^\top(P_{K'} - P_K)A + B^\top(P_{K'} - P_K)BK' + B^\top P_K B(K' - K).$$

Since $\|K'\| \le 2\|K\|$, and $\|K\|$ can be bounded by $C(K)$ (Lemma 19), all the terms can be bounded by polynomials of related parameters multiplied by $\|K - K'\|$. $\qquad\square$

### D.3    SMOOTHING AND THE GRADIENT DESCENT ANALYSIS

This section analyzes the smoothing procedure and completes the proof of gradient descent. Although Gaussian smoothing is more standard, the objective $C(K)$ is not finite for every $K$, therefore technically $\mathbb{E}_{u\sim\mathcal{N}(0,\sigma^2 I)}[C(K + u)]$ is not well defined. This is avoidable by smoothing in a ball.

Let $\mathbb{S}_r$ represent the uniform distribution over the points with norm $r$ (boundary of a sphere), and $\mathbb{B}_r$ represent the uniform distribution over all points with norm at most $r$ (the entire sphere). When applying these sets to matrix a $U$, the Frobenius norm ball is used. The algorithm performs gradient descent on the following function

$$C_r(K) = \mathbb{E}_{U\sim\mathbb{B}_r}[C(K + U)].$$

The next lemma uses the standard technique (e.g. in (Flaxman et al., 2005)) to show that the gradient of $C_r(K)$ can be estimated just with an oracle for function value.

**Lemma 23.** $\nabla C_r(K) = \frac{d}{r^2}\mathbb{E}_{U\sim\mathbb{S}_r}[C(K + U)U]$.

This is the same as Lemma 2.1 in Flaxman et al. (2005), for completeness the proof is provided below.

*Proof.* By Stokes formula,

$$\nabla \int_{\delta\mathbb{B}_r} C(K + U)dx = \int_{\delta\mathcal{S}_r} C(K + u)\frac{U}{\|U\|_F}dx.$$

By definition,

$$C_r(K) = \frac{\int_{\delta\mathbb{B}_r} C(K + U)dx}{\mathrm{vol}_d(\delta\mathbb{B}_r)},$$

Also,

$$\mathbb{E}_{U \sim \mathbb{S}_r}[C(K+U)U] = r\mathbb{E}_{U \sim \mathbb{S}_r}[C(K+U)\frac{U}{r}] = r \cdot \frac{\int_{\delta \mathcal{S}_r} C(K+U)\frac{U}{\|U\|_F}dx}{\mathrm{vol}_{d-1}(\delta \mathbb{S}_r)}.$$

The Lemma follows from combining these equations, and use the fact that

$$\mathrm{vol}_d(\delta \mathbb{B}_r) = \mathrm{vol}_{d-1}(\delta \mathbb{S}_r) \cdot \frac{r}{d}.$$

$\square$

From the lemma above and standard concentration inequalities, it is immediate that it suffices to use a polynomial number of samples to approximate the gradient.

**Lemma 24.** *Given an $\epsilon$, there are fixed polynomials $h_r(1/\epsilon), h_{sample}(d, 1/\epsilon)$ such that when $r \le 1/h_r(1/\epsilon)$, with $m \ge h_{sample}(d, 1/\epsilon)$ samples of $U_1, ..., U_n \sim \mathbb{S}_r$, with high probability (at least $1 - (d/\epsilon)^{-d}$) the average*

$$\hat{\nabla} = \frac{1}{m}\sum_{i=1}^m \frac{d}{r^2}C(K+U_i)U_i$$

*is $\epsilon$ close to $\nabla C(K)$.*

*Further, if for $x \sim \mathcal{D}$, $\|x\| \le L$ almost surely, there are polynomials $h_{\ell,grad}(d, 1/\epsilon)$, $h_{r,trunc}(1/\epsilon)$, $h_{sample,trunc}(d, 1/\epsilon, \sigma, L^2/\mu)$ such that when $m \ge h_{sample,trunc}(d, 1/\epsilon, L^2/\mu)$, $\ell \ge h_{\ell,grad}(d, 1/\epsilon)$, let $x_j^i, u_j^i(0 \le j \le \ell)$ be a single path sampled using $K + U_i$, then the average*

$$\tilde{\nabla} = \frac{1}{m}\sum_{i=1}^m \frac{d}{r^2}[\sum_{j=0}^{\ell-1}(x_j^i)^\top Q x_j^i + (u_j^i)^\top R u_j^i]U_i$$

*is also $\epsilon$ close to $\nabla C(K)$ with high probability.*

*Proof.* For the first part, the difference is broken into two terms:

$$\hat{\nabla} - \nabla C(K) = (\nabla C_r(K) - \nabla C(K)) + (\hat{\nabla} - \nabla C_r(K)).$$

For the first term, choose $h_r(1/\epsilon) = \min\{1/r_0, 2h_{grad}/\epsilon\}$ ($r_0$ is chosen later). By Lemma 22 when $r$ is smaller than $1/h_r(1/\epsilon) = \epsilon/2h_{grad}$, every point $u$ on the sphere have $\|\nabla C(K+U) - \nabla C(K)\| \le \epsilon/4$. Since $\nabla C_r(K)$ is the expectation of $\nabla C(K+U)$, by triangle inequality $\|\nabla C_r(K) - \nabla C(K)\| \le \epsilon/2$.

The proof also makes sure that $r \le r_0$ such that for any $U \sim \mathbb{S}_r$, it holds that $C(K+U) \le 2C(K)$. By Lemma 21, $1/r_0$ is a polynomial in the relevant factors.

For the second term, by Lemma 23, $\mathbb{E}[\hat{\nabla}] = \nabla C_r(K)$, and each individual sample has norm bounded by $2dC(K)/r$, so by Vector Bernstein's Inequality, know with $m \ge h_{sample}(d, 1/\epsilon) = \Theta\left(d\left(\frac{dC(K)}{\epsilon r}\right)^2\right)\log d/\epsilon\right)$ samples, with high probability (at least $1 - (d/\epsilon)^{-d}$) $\|\hat{\nabla} - \mathbb{E}[\hat{\nabla}]\| \le \epsilon/2$.

Adding these two terms and apply triangle inequality gives the result.

For the second part, the proof breaks it into more terms. Let $\nabla'$ be equal to $\frac{1}{m}\sum_{i=1}^m \frac{d}{r^2}C^{(\ell)}(K+U_i)U_i$ (where $C^{(\ell)}$ is defined as in Lemma 20), then

$$\tilde{\Sigma} - \nabla C(K) = (\tilde{\Sigma} - \Sigma') + (\Sigma' - \hat{\Sigma}) + (\hat{\Sigma} - \nabla C(K)).$$

The third term is just what was bounded earlier, using $h_{r,trunc}(1/\epsilon) = h_r(2/\epsilon)$ and making sure $h_{sample,trunc}(d, 1/\epsilon) \ge h_{sample}(d, 2/\epsilon)$, this guarantees that it is smaller than $\epsilon/2$.

For the second term, choose $\ell \ge \frac{16d^2 \cdot C^2(K)(\|Q\| + \|R\|\|K\|^2)}{\epsilon r \mu \sigma_{min}^2(Q)} =: h_{\ell,grad}(d, 1/\epsilon)$. By Lemma 20, for any $K'$ with $C(K') \le 2C(K)$, it holds that $\|C^{(\ell)}(K') - C(K')\| \le \frac{r\epsilon}{4d}$. Therefore by triangle inequality

$$\|\frac{1}{m}\sum_{i=1}^m \frac{d}{r^2}C^{(\ell)}(K+U_i)U_i - \frac{1}{m}\sum_{i=1}^m \frac{d}{r^2}C(K+U_i)U_i\| \le \epsilon/4.$$

Finally for the first term it is easy to see that $\mathbb{E}[\tilde{\nabla}] = \nabla'$ where the expectation is taken over the randomness of the initial states $x_0^i$. Since $\|x_0^i\| \le L$, $(x_0^i)(x_0^i)^\top \preceq \frac{L^2}{\mu}\mathbb{E}[x_0 x_0^\top]$, as a result the sum

$$[\sum_{j=0}^{\ell-1}(x_j^i)^\top Q x_j^i + (u_j^i)^\top R u_j^i] \le \frac{L^2}{\mu}C(K + U_i).$$

Therefore, $\tilde{\nabla} - \nabla'$ is again a sum of independent vectors with bounded norm, so by Vector Bernstein's inequality, when $h_{sample,trunc}(d, 1/\epsilon, L^2/\mu)$ is a large enough polynomial, $\|\tilde{\nabla} - \nabla'\| \le \epsilon/4$ with high probability. Adding all the terms finishes the proof. $\square$

**Theorem 25.** *There are fixed polynomials $h_{GD,r}(1/\epsilon), h_{GD,sample}(d, 1/\epsilon, L^2/\mu), h_{GD,\ell}(d, 1/\epsilon)$ such that if every step the gradient is computed as Lemma 24 (truncated at step $\ell$), pick step size $\eta$ and $T$ the same as the gradient descent case of Theorem 5, it holds that $C(K_T) - C(K^\star) \le \epsilon$ with high probability (at least $1 - \exp(-d)$).*

*Proof.* By Lemma 18, when $\eta \le 1/h_{GD,\eta}$ for some fixed polynomial $h_{GD,\eta}$(given in Lemma 18), then

$$C(K') - C(K^*) \le \left(1 - \eta\sigma_{\min}(R)\frac{\mu^2}{\|\Sigma_{K^*}\|}\right)(C(K) - C(K^*))$$

Let $\tilde{\nabla}$ be the approximate gradient computed, and let $K'' = K - \eta\tilde{\nabla}$ be the iterate that uses the approximate gradient. The proof shows given enough samples, the gradient can be estimated with enough accuracy that makes sure

$$|C(K'') - C(K')| \le \frac{1}{2}\eta\sigma_{\min}(R)\frac{\mu^2}{\|\Sigma_{K^*}\|} \cdot \epsilon.$$

This means as long as $C(K) - C(K^*) \ge \epsilon$, it holds that

$$C(K'') - C(K^*) \le \left(1 - \frac{1}{2}\eta\sigma_{\min}(R)\frac{\mu^2}{\|\Sigma_{K^*}\|}\right)(C(K) - C(K^*)).$$

Then the same proof of Theorem 5 gives the convergence guarantee.

Now $C(K'') - C(K')$ is bounded. By Lemma 21, if $\|K'' - K'\| \le \frac{1}{2}\eta\sigma_{\min}(R)\frac{\mu^2}{\|\Sigma_{K^*}\|} \cdot \epsilon \cdot 1/h_{func}$ ($h_{func}$ is the polynomial in Lemma 21), then $C(K'') - C(K')$ is small enough. To get that, observe $K'' - K' = \eta(\nabla - \tilde{\nabla})$, therefore it suffices to make sure

$$\|\nabla - \tilde{\nabla}\| \le \frac{1}{2}\sigma_{\min}(R)\frac{\mu^2}{\|\Sigma_{K^*}\|} \cdot \epsilon \cdot 1/h_{func}$$

By Lemma 22, it suffices to pick $h_{GD,r}(1/\epsilon) = h_{r,trunc}(2h_{func}\|\Sigma_{K^*}\|/(\mu^2\sigma_{min}(R)\epsilon))$, $h_{GD,sample}(d, 1/\epsilon, L^2/\mu) = h_{sample,trunc}(d, 2h_{func}\|\Sigma_{K^*}\|/(\mu^2\sigma_{min}(R)\epsilon), L^2/\mu)$, and $h_{GD,\ell}(d, 1/\epsilon) = h_{\ell,grad}(d, 2h_{func}\|\Sigma_{K^*}\|/(\mu^2\sigma_{min}(R)\epsilon))$. This gives the desired upper-bound on $\|\nabla - \tilde{\nabla}\|$ with high probability (at least $1 - (\epsilon/d)^{-d}$).

Since the number of steps is a polynomial, by union bound with high probability (at least $1 - T(\epsilon/d)^{-d} \ge 1 - \exp(-d)$) the gradient is accurate enough for all the steps, so

$$C(K'') - C(K^*) \le \left(1 - \frac{1}{2}\eta\sigma_{\min}(R)\frac{\mu^2}{\|\Sigma_{K^*}\|}\right)(C(K) - C(K^*)).$$

The rest of the proof is the same as Theorem 5. Note that in the smoothing, because the function value is monotonically decreasing and the choice of radius, all the function value encountered is bounded by $2C(K_0)$, so the polynomials are indeed bounded throughout the algorithm. $\square$

### D.4 THE NATURAL GRADIENT ANALYSIS

Before the Theorem for natural gradient is proven, the following lemma shows the variance can be estimated accurately.

**Lemma 26.** *If for $x \sim \mathcal{D}$, $\|x\| \leq L$ almost surely, there exists polynomials $h_{r,var}(1/\epsilon)$, $h_{varsample,trunc}(d, 1/\epsilon, L^2/\mu)$ and $h_{\ell,var}(d, 1/\epsilon)$ such that if $\hat{\Sigma}_K$ is estimated using at least $m \geq h_{varsample,trunc}(d, 1/\epsilon, L^2/\mu)$ initial points $x_0^1, ..., x_0^m$, $m$ random perturbations $U_i \sim \mathbb{S}_r$ where $r \leq 1/h_{r,var}(1/\epsilon)$, all of these initial points are simulated using $\hat{K}_i = K + U_i$ to $\ell \geq h_{\ell,var}(d, 1/\epsilon)$ iterations, then with high probability (at least $1 - (d/\epsilon)^{-d}$) the following estimate*

$$\tilde{\Sigma} = \frac{1}{m} \sum_{i=1}^{m} \sum_{j=0}^{\ell-1} x_j^i (x_j^i)^\top.$$

*satisfies $\|\tilde{\Sigma} - \Sigma_K\| \leq \epsilon$. Further, when $\epsilon \leq \mu/2$, it holds that $\sigma_{min}(\hat{\Sigma}_K) \geq \mu/2$.*

*Proof.* This is broken into three terms: let $\Sigma_K^{(\ell)}$ be defined as in Lemma 20, let $\hat{\Sigma} = \frac{1}{m} \sum_{i=1}^{m} \Sigma_{K+U_i}$ and $\hat{\Sigma}^{(\ell)} = \frac{1}{m} \sum_{i=1}^{m} \Sigma_{K+U_i}^{(\ell)}$, then it holds that

$$\tilde{\Sigma} - \Sigma_K = (\tilde{\Sigma} - \hat{\Sigma}^{(\ell)}) + (\hat{\Sigma}^{(\ell)} - \hat{\Sigma}) + (\hat{\Sigma} - \Sigma_K).$$

First, $r$ is chosen small enough so that $C(K + U_i) \leq 2C(K)$. This only requires an inverse polynomial $r$ by Lemma 21.

For the first term, note that $\mathbb{E}[\tilde{\Sigma}] = \hat{\Sigma}^{(\ell)}$ where the expectation is taken over the initial points $x_0^i$. Since $\|x_0^i\| \leq L$, $(x_0^i)(x_0^i)^\top \preceq \frac{L^2}{\mu} \mathbb{E}[x_0 x_0^\top]$, and as a result the sum

$$\sum_{j=0}^{\ell-1} x_j^i (x_j^i)^\top Q \preceq \frac{L^2}{\mu} \Sigma_{K+U_i}.$$

Therefore, standard concentration bounds show that when $h_{varsample,trunc}$ is a large enough polynomial, $\|\tilde{\Sigma} - \hat{\Sigma}^{(\ell)}\| \leq \epsilon/2$ holds with high probability.

For the second term, Lemma 20 is applied. Because $C(K + U_i) \leq 2C(K)$, choosing $\ell \geq h_{\ell,var}(d, 1/\epsilon) = \frac{8d \cdot C^2(K)}{\epsilon \mu \sigma_{min}^2(Q)}$, the error introduced by truncation $\|\hat{\Sigma}^{(\ell)} - \hat{\Sigma}\|$ is then bounded by $\epsilon/4$.

For the third term, Lemma 13 is applied. When $r \leq \epsilon \cdot \left(\frac{\sigma_{min}(Q)}{C(K)}\right)^2 \frac{\mu}{16\|B\|(\|A-BK\|+1)}$, $\|\Sigma_{K+U_i} - \Sigma_K\| \leq \epsilon/4$. Since $\hat{\Sigma}$ is the average of $\Sigma_{K+U_i}$, by the triangle inequality, $\|\hat{\Sigma} - \Sigma_K\| \leq \epsilon/4$.

Adding these three terms gives the result.

Finally, the bound on $\sigma_{min}(\tilde{\Sigma}_K)$ follows simply from Weyl's Theorem. $\square$

**Theorem 27.** *Suppose $C(K_0)$ is finite and and $\mu > 0$. The natural gradient follows the update rule:*

$$K_{t+1} = K_t - \eta \nabla C(K_t) \Sigma_{K_t}^{-1}$$

*Suppose the stepsize is set to be:*

$$\eta = \frac{1}{\|R\| + \frac{\|B\|^2 C(K_0)}{\mu}}$$

*If the gradient and variance are estimated as in Lemma 24, Lemma 26 with $r = 1/h_{NGD,r}(1/\epsilon)$, with $m \geq h_{NGD,sample}(d, 1/\epsilon, L^2/\mu)$ samples, both are truncated to $h_{NGD,\ell}(d, 1/\epsilon)$ iterations, then with high probability (at least $1 - \exp(-d)$) in $T$ iterations where*

$$T > \frac{\|\Sigma_{K^*}\|}{\mu} \left(\frac{\|R\|}{\sigma_{min}(R)} + \frac{\|B\|^2 C(K_0)}{\mu \sigma_{min}(R)}\right) \log \frac{2(C(K_0) - C(K^*))}{\varepsilon}$$

*then the natural gradient satisfies the following performance bound:*

$$C(K_T) - C(K^*) \leq \varepsilon$$

*Proof.* By Lemma 12,

$$C(K') - C(K^*) \le \left(1 - \eta\sigma_{\min}(R)\frac{\mu}{\|\Sigma_{K^*}\|}\right)(C(K) - C(K^*))$$

Let $\tilde{\nabla}$ be the estimated gradient, $\tilde{\Sigma}_K$ be the estimated $\Sigma_K$, and let $K'' = K - \eta\tilde{\nabla}\tilde{\Sigma}_K^{-1}$. The proof shows that when both the gradient and the covariance matrix are estimated accurately enough, then

$$|C(K') - C(K'')| \le \frac{\epsilon}{2}\eta\sigma_{\min}(R)\frac{\mu}{\|\Sigma_{K^*}\|}.$$

This implies when $C(K) - C(K^\star) \ge \epsilon$,

$$C(K') - C(K^*) \le \left(1 - \frac{1}{2}\eta\sigma_{\min}(R)\frac{\mu}{\|\Sigma_{K^*}\|}\right)(C(K) - C(K^*))$$

which is sufficient for the proof.

By Lemma 21, if $\|K'' - K'\| \le \frac{\epsilon}{2h_{func}}\eta\sigma_{\min}(R)\frac{\mu}{\|\Sigma_{K^*}\|}$ the desired bound on $|C(K') - C(K'')|$ holds. To achieve this, it suffices to have

$$\|\tilde{\nabla}\tilde{\Sigma}_K^{-1} - \nabla C(K)\Sigma_K^{-1}\| \le \frac{\epsilon}{2h_{func}}\sigma_{\min}(R)\frac{\mu}{\|\Sigma_{K^*}\|}.$$

This is broken into two terms

$$\|\tilde{\nabla}\tilde{\Sigma}_K^{-1} - \nabla C(K)\Sigma_K^{-1}\| \le \|\tilde{\nabla} - \nabla\|\|\tilde{\Sigma}_K^{-1}\| + \|\nabla C(K)\|\|\tilde{\Sigma}_K^{-1} - \Sigma_K^{-1}\|.$$

For the first term, by Lemma 26 we know when the number of samples is large enough $\|\tilde{\Sigma}_K^{-1}\| \le 2/\mu$. Therefore it suffices to make sure $\|\tilde{\nabla} - \nabla\| \le \frac{\epsilon}{8h_{func}}\sigma_{\min}(R)\frac{\mu^2}{\|\Sigma_{K^*}\|}$, this can be done by Lemma 24 by setting $h_{NGD,grad,r}(1/\epsilon) = h_{r,trunc}(\frac{8h_{func}\|\Sigma_K^*\|}{\mu^2\sigma_{min}(R)\epsilon})$, $h_{NGD,gradsample}(d,1/\epsilon,L/\mu^2) = h_{sample,trunc}(d, \frac{8h_{func}\|\Sigma_K^*\|}{\mu^2\sigma_{min}(R)\epsilon}, L/\mu^2)$ and $h_{NGD,\ell,grad}(d,1/\epsilon) = h_{\ell,grad}(d, \frac{8h_{func}\|\Sigma_K^*\|}{\mu^2\sigma_{min}(R)\epsilon})$.

For the second term, it suffices to make sure $\|\tilde{\Sigma}_K^{-1} - \Sigma_K^{-1}\| \le \frac{\epsilon}{4h_{func}}\sigma_{\min}(R)\frac{\mu}{\|\Sigma_{K^*}\|\|\nabla C(K)\|}\cdots$ By standard matrix perturbation, and $\sigma_{min}(\Sigma_K) \ge \mu$, $\|\tilde{\Sigma}_K^{-1} - \Sigma_K^{-1}\| \le 2\|\tilde{\Sigma}_K - \Sigma_K\|/\mu^2$ (when $\|\tilde{\Sigma}_K - \Sigma_K\| \le \mu/2$). Therefore by Lemma 26 it suffices to choose $h_{NGD,var,r}(1/\epsilon) = h_{var,r}(\frac{8h_{func}\|\Sigma_{K^*}\|\|\nabla C(K)\|}{\mu^3\sigma_{min}(R)\epsilon})$, $h_{NGD,varsample}(d,1/\epsilon,L/\mu^2) = h_{varsample,trunc}(d, \frac{8h_{func}\|\Sigma_{K^*}\|\|\nabla C(K)\|}{\mu^3\sigma_{min}(R)\epsilon}, L/\mu^2)$ and $h_{NGD,\ell,var}(d,1/\epsilon) = h_{\ell,var}(d, \frac{8h_{func}\|\Sigma_{K^*}\|\|\nabla C(K)\|}{\mu^3\sigma_{min}(R)\epsilon})$. This is indeed a polynomial because $\|\nabla C(K)\|$ is bounded by Lemma 19.

Finally, choose $h_{NGD,r} = \max\{h_{NGD,grad,r}, h_{NGD,var,r}\}$, $h_{NGD,sample} = \max\{h_{NGD,gradsample}, h_{NGD,varsample}\}$, and $h_{NGD,\ell} = \max\{h_{NGD,\ell,grad}, h_{NGD,\ell,var}\}$. This ensures all the bounds mentioned above hold and that

$$C(K') - C(K^*) \le \left(1 - \frac{1}{2}\eta\sigma_{\min}(R)\frac{\mu}{\|\Sigma_{K^*}\|}\right)(C(K) - C(K^*))$$

The rest of the proof is the same as Theorem 5. Note again that in the smoothing, because the function value is monotonically decreasing and the choice of radius, all the function values encountered are bounded by $2C(K_0)$, so the polynomials are indeed bounded throughout the algorithm. $\square$

