# OpenReview forum: "Global Convergence of Policy Gradient Methods for Linearized  Control Problems"
_ICLR.cc/2018/Conference — Reject_

### Official Review · AnonReviewer2 · 2017-11-27
**GLOBAL CONVERGENCE OF POLICY GRADIENT METHODS FOR LINEARIZED CONTROL PROBLEMS**

**Rating:** 6
**Confidence:** 4

**Review:**

The work investigates convergence guarantees of gradient-type policies for reinforcement learning and continuous control
problems, both in deterministic and randomized case, whiling coping with non-convexity of the objective. I found that the paper suffers many shortcomings that must be addressed:

1) The writing and organization is quite cumbersome and should be improved.
2) The authors state in the abstract (and elsewhere): "... showing that (model free) policy gradient methods globally converge to the optimal solution ...". This is misleading and NOT true. The authors show the convergence of the objective but not of the iterates sequence. This should be rephrased elsewhere.
3) An important literature on convergence of descent-type methods for semialgebraic objectives is available but not discussed.

---

> ### Author Response · Authors · 2017-12-22
> **author response**
>
> 1. What we mean by 'rate of convergence’: as is clear in our theorems, we aim to show convergence rates for the objective value. This is standard in optimization literature. To be more clear and explicit in the abstract and introduction, we will update to say that "the algorithms converge to a controller K with objective that's epsilon-close to optimal value." We could also prove the convergence of the iterate sequence (the parameters) but that is not our main interest.
>
> 2. Perhaps what the reviewer is referring to is the literature behind Kurdyka-Lojasiewicz (KL) or Polyak-Lojasiewicz (PL) inequalities and what functions satisfy them---often functions satisfy these properties with a known exponent only *locally* (e.g., families of semialgebraic functions) and then the inequalities are used to show rates of convergence to a stationary point.
> We are also using the KL inequality, but what's interesting is that for us the KL inequality hold globally (the only assumption is for C(K0) to be bounded) and we are able to show convergence (in function values) to the globally optimal value. This kind of situation is rare, we haven't seen too many nontrivial functions satisfying it.We believe this is an interesting new viewpoint for LQR that the controls community has not taken before.

---

### Official Review · AnonReviewer1 · 2017-11-27

**Rating:** 5
**Confidence:** 3

**Review:**

I find this paper not suitable for ICLR. All the results are more or less direct applications of existing optimization techniques, and not provide fundamental new understandings of the learning REPRESENTATION.

---

> ### Author Response · Authors · 2017-12-22
> **author response**
>
> 1. This paper proves the convergence of several algorithms (policy gradient, natural policy gradient) that are widely used in recent developments in reinforcement learning (including the ones using neural networks and learning representations). We believe understanding the behavior of these algorithms in the LQR setting is an important and necessary first step before understanding the more complicated neural network settings. Also, a common technique in practice is to approximate the problem locally as linear dynamical systems, and our results can be applied in these settings.
> 2. Our result are not direct applications of existing optimization techniques. As we observed in the paper, the problem is non-convex (and is not even quasi-convex or star-convex) and existing techniques do not work in this setting. We do draw analogs to some familiar concepts (such as smoothness) in optimization, but the way we prove these guarantees is very different from the existing literature.

---

### Official Review · AnonReviewer3 · 2017-12-12
**Global Convergence of Policy Gradient Methods for Linearized Control Problems**

**Rating:** 5
**Confidence:** 3

**Review:**

The paper studies the global convergence for policy gradient methods for linear control problems.
(1) The topic of this paper seems to have minimal connection with ICRL. It might be more appropriate for this paper to be reviewed at a control/optimization conference, so that all the technical analysis can be evaluated carefully.

(2) I am not convinced if the main results are novel. The convergence of policy gradient does not rely on the convexity of the loss function, which is known in the community of control and dynamic programming. The convergence of policy gradient is related to the convergence of actor-critic, which is essentially a form of policy iteration. I am not sure if it is a good idea to examine the convergence purely from an optimization perspective.

(3) The main results of this paper seem technical sound. However, the results seem a bit limited because it does not apply to neural-network function approximator. It does not apply to the more general control problem rather than quadratic cost function, which is quite restricted. I might have missed something here. I strongly suggest that these results be submitted to a more suitable venue.

---

> ### Author Response · Authors · 2017-12-22
> **author response**
>
> 1. This paper proves the convergence of several algorithms (policy gradient, natural policy gradient) that are widely used in the recent developments in reinforcement learning (including the ones using neural networks and learning representations). We believe understanding the behavior of these algorithms in the LQR setting is an important and necessary first step before understanding the more complicated neural network settings. Also, a common technique in practice is to approximate the problem locally as linear dynamical systems, and our results can be applied in these settings.
> 2. We are not aware of any global convergence guarantees in the general non-convex setting. There are some convergence guarantees in convex settings, but even in convex settings there are worst-case examples that require super-polynomial number of iterations for policy iteration (for example, the construction in paper “Sub-exponential lower bounds for randomized pivoting rules for solving linear programs” by Friedmann et al.). In the general non-convex setting, even converge to a local minimum (rather than a saddle point) can take exponential time. Our contribution is to prove that policy gradient actually converges in polynomial number of iterations in the setting of LQR.

---

### Decision · Program_Chairs · 2018-01-29
**ICLR 2018 Conference Acceptance Decision**

**Decision:**

Reject

**Comment:**

The paper studies the global convergence for policy gradient methods for linear control problems.  Multiple reviewers point out strong concerns about the novelty of the results.